# NECOMIMI: Neural-Cognitive Multimodal EEG-informed Image Generation with Diffusion Models

## Abstract

NECOMIMI (NEural-COgnitive MultImodal EEG-Informed Image Generation with Diffusion Models) introduces a novel framework for generating images directly from EEG signals using advanced diffusion models. Unlike previous works that focused solely on EEG-image classification through contrastive learning, NECOMIMI extends this task to image generation. The proposed NERV EEG encoder demonstrates state-of-the-art (SoTA) performance across multiple zero-shot classification tasks, including 2-way, 4-way, and 200-way, and achieves top results in our newly proposed CAT Score, which evaluates the quality of EEG-generated images based on semantic concepts. A key discovery of this work is that the model tends to generate abstract or generalized images, such as landscapes, rather than specific objects, highlighting the inherent challenges of translating noisy and low-resolution EEG data into detailed visual outputs. Additionally, we introduce the CAT Score as a new metric tailored for EEG-to-image evaluation and establish a benchmark on the ThingsEEG dataset. This study underscores the potential of EEG-to-image generation while revealing the complexities and challenges that remain in bridging neural activity with visual representation.

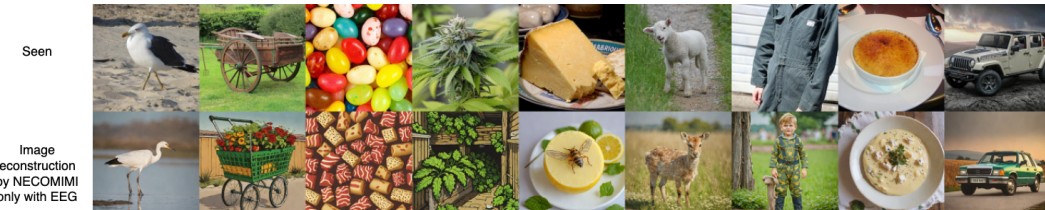

Figure 1: This image demonstrates the capability of the NECOMIMI model to reconstruct images purely from EEG data without using the "Seen" images (ground truth) as embeddings during the generation process. The two-stage NECOMIMI architecture effectively extracts semantic information from noisy EEG signals, showing that it can capture and represent the underlying concepts from brainwave activity. The bottom row of images, generated solely from EEG input, highlights the potential of NECOMIMI to approximate the content of the "Seen" images in the top row, even in the absence of any direct visual reference or embedding.

## 1 Introduction

Electroencephalography (EEG) is one of the most ancient techniques used to measure neuronal activity in the human brain Mary (1959); Millett (2001). Its application has significant value in clinical practice, particularly in diagnosing epilepsy Reif et al. (2016), depression Li et al. (2023) and sleep disorders Hussain et al. (2022), as well as in assessing dysfunctions in sensory transmission pathways Thoma et al. (2003) and more Perrottelli et al. (2021). Historically, the analysis of EEG signals was limited to visual inspection of amplitude and frequency changes over time. However, with advancements in digital technology, the methodology has evolved significantly, shifting towards a more comprehensive analysis of the temporal and spatial characteristics of these signals EK;Frey

(2016). As a result of this evolution, EEG has gained recognition as a potent tool for capturing brain functions in real-time, particularly in the sub-second range. Despite its advantages, EEG has traditionally suffered from poor spatial resolution, making it challenging to pinpoint the precise brain areas responsible for the measured neuronal activity at the scalp Li et al. (2022). In recent years, there has been a surge of interest in utilizing EEG for more sophisticated applications, such as image recognition and reconstruction Mai et al. (2023). These advancements have led to significant improvements in the accuracy of image recognition tasks, underscoring the potential of EEG as a bridge between neural activity and visual representation Spampinato et al. (2016); Kavasidis et al. (2017). The growing interest in using EEG for image recognition is rooted in its ability to capture the temporal dynamics of neuronal activity, though its spatial resolution remains a challenge. Innovative methodologies, including deep learning techniques and generative models like Generative Adversarial Networks (GANs) Goodfellow et al. (2014) and diffusion models Ho et al. (2020), have enhanced the accuracy and effectiveness of EEG-based systems, allowing for the generation of photorealistic images based on neural signals Kavasidis et al. (2017); Kumar et al. (2017); Singh et al. (2023). Notably, studies have demonstrated the feasibility of decoding natural images from EEG signals, employing innovative frameworks that align EEG responses with paired image stimuli Bai et al. (2023). However, most of the current works claiming to be EEG-to-image are essentially still image-to-image in nature, with EEG information primarily used to slightly guide the transformation of the input image by adding noise Kavasidis et al. (2017); Palazzo et al. (2017); Khare et al. (2022); Bai et al. (2023). In order to achieve a truly meaningful EEG-to-image generation, this work, named NECOMIMI (NEural-COgnitive MultImodal eeg-inforMed Image generation with diffusion models), introduces an innovative framework focused on EEG-based image generation, combining advanced diffusion model techniques.

This paper presents several key innovations as follows:

- We propose a novel EEG encoder, NERV, which achieves state-of-the-art performance in multimodal contrastive learning tasks.

- Unlike previous work that primarily focused on image-to-image generation with EEG features as guidance, we introduce a comprehensive two-stage EEG-to-image multimodal generative framework. This not only extends prior contrastive learning between EEG and images but also applies it to image generation.

- To address the conceptual differences between EEG-to-image and traditional text-to-image tasks, we propose a new quantification method, the Category-based Assessment Table (CAT) Score, which evaluates image generation performance based on semantic concepts rather than image distribution.

- We establish a CAT score benchmark standard using Vision Language Model (VLM) on the ThingsEEG dataset.

- Additionally, we uncover some notable findings and phenomena regarding the EEG-to-image generation process.

## 2 RELATED WORKS

### 2.1 THE POTENTIAL OF EEG DATA

In a typical experiment studying brain responses related to visual processes, a person looks at a series of images while a brain scanner or recording device captures their brain signals for analysis. There are various non-invasive methods to capture these brain responses, like fMRI, EEG, and MEG, each with different sensitivity levels. However, we still don't fully understand what this data really means, and even more importantly, how to interpret it. In a pioneering study Nishimoto et al. (2011), the researchers tried to generate impressions of what the subjects saw using fMRI images, based on a large image dataset taken from YouTube. However, this method has challenges, like the complexity and high cost of using an fMRI scanner. To overcome these drawbacks, a lot of research has shifted to using electrophysiological responses, particularly EEG, which has lower spatial resolution than most other methods but much higher temporal resolution. EEG recordings are also cheaper and easier to conduct, but the data is often noisy and affected by external factors, making it harder to reconstruct the original stimulus. Most image recognition and/or generation from brain signals nowadays is done using fMRI data Zhang et al. (2023), while EEG, being noisier, is used much less often.

## 2.2 USING EEG INFORMATION ON IMAGE GENERATION AND RECONSTRUCTION

Building on this shift towards EEG, prior to efforts in generating images directly from brain data, the concept of using EEG signals for image classification was introduced by the study Spampinato et al. (2017). This work first demonstrated the feasibility of decoding visual categories from EEG recordings using deep learning models, setting a foundation for leveraging neural signals in image-related tasks. However, the dataset they used was relatively small, which limited the generalization of their findings. Further advancements in generative models, specifically with the introduction of Variational Autoencoders (VAE) and Generative Adversarial Networks (GAN), opened new possibilities for image generation. The VAE model proposed by Kingma & Welling (2013; 2019) achieved data generation and reconstruction by learning the latent distribution of data. The GAN model introduced by Goodfellow et al. (2014) utilized adversarial training between a generator and a discriminator to produce highly realistic images. Building on these methods, Brain2Image Kavasidis et al. (2017) was the first to use VAE to guide image generation from EEG features. Following that, EEG-GAN Palazzo et al. (2017) presented the first EEG-based image generation model, using LSTM Hochreiter & Schmidhuber (1997) to extract EEG information and guide the GAN for image generation. After this, there were still many EEG-to-image works based on GAN that emerged, with most of them focusing on improving the GAN architecture and the way it interacts with the EEG encoder, like in ThoughtViz Tirupattur et al. (2018), VG-GAN-VC Jiao et al. (2019), BrainMedia Fares et al. (2020), and EEG2IMAGE Singh et al. (2023), etc. However, in all these works, a common and challenging problem is figuring out how to effectively use EEG data to guide image generation and reconstruction. This challenge of training neural networks to align multimodal information wasn't effectively addressed until the emergence of CLIP Radford et al. (2021a), which provided a much better solution. Since then, some works have also applied this approach to EEG-based image generation.

## 2.3 CONTRASTIVE LEARNING-BASED WORKS ON EEG-IMAGE TASKS

To the best of our knowledge, EEGCLIP Singh et al. (2024) was the first to use contrastive learning to align EEG and image data. However, in this work, this aspect was only an exploratory attempt and did not further utilize the framework for downstream tasks like zero-shot image recognition. The next challenge lies in designing a better EEG encoder for contrastive learning, based on the rich image embeddings extracted from a CLIP-based image pre-trained encoder. Some recent works have explored this direction, such as NICE Song et al. (2024), MUSE Chen & Wei (2024), ATM Li et al. (2024), and Chen et al. (2024c). Some researchers have even attempted quantum-classical hybrid computing and quantum EEG encoder Chen et al. (2024a) to perform quantum contrastive learning Chen et al. (2024b). Most current works focus on tackling zero-shot classification, where the model is tested on unseen both EEG data and images that it hasn't encountered during training. The goal is to compute similarity scores for image recognition, aiming to enhance the model's generalization performance on out-of-sample data. As contrastive learning architectures for EEG-based image recognition mature, and inspired by test-to-image frameworks in other generative fields, the invention of diffusion models has addressed the instability issues associated with previous GAN-based generation methods to some extent. While there are already EEG-based image reconstruction efforts using diffusion models, such as NeuroVision Khare et al. (2022), DreamDiffusion Bai et al. (2023), DM-RE2I Zeng et al. (2023), BrainViz Fu et al. (2023), NeuroImagen Lan et al. (2023), and EEGVision Guo (2024), most of these works still largely rely on image-based features, with EEG data serving as supplementary information for the diffusion process. While these methods have made significant strides in computer vision, they primarily rely on images as input and are not designed to process non-visual signals like EEG directly. Currently, models designed specifically for direct generation tasks using pure EEG features or embeddings, where EEG functions similarly to a prompt command, are still quite rare. This work seeks to introduce a flexible, plug-and-play architecture: NECOMIMI, which not only expands upon previous recognition-focused approaches but also extends them into EEG-to-image generation tasks based on modern diffusion models.

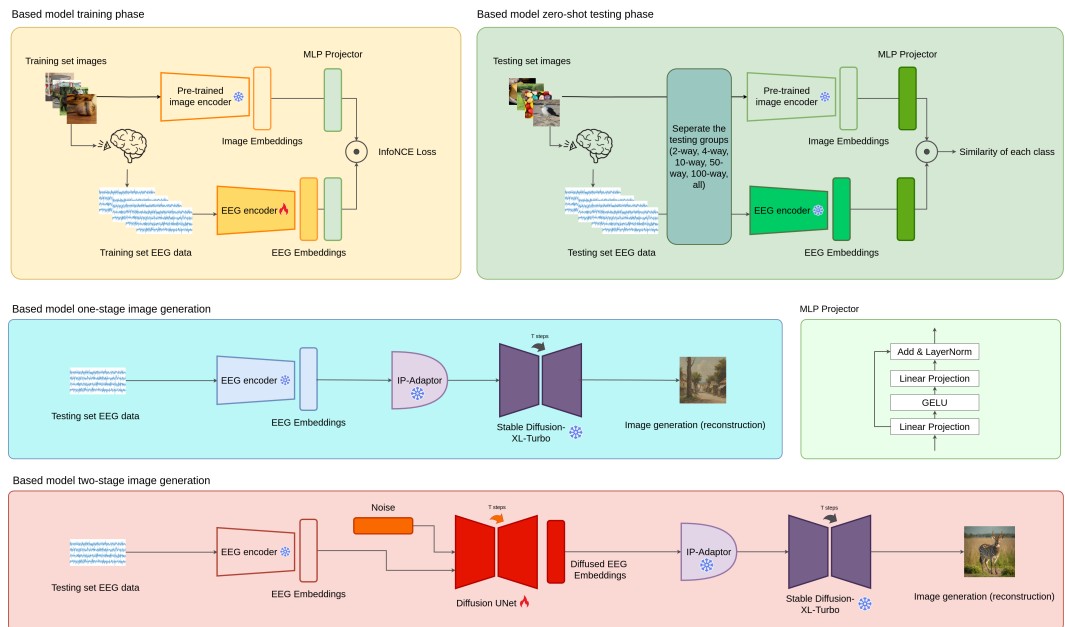

Figure 2: The figure illustrates the entire workflow of the EEG-based image generation model.

# 3 METHODOLOGY

## 3.1 OVERVIEW

This chapter provides a detailed overview of an advanced EEG-to-image generation model utilizing deep learning techniques and diffusion models. While the framework includes a one-stage image generation phase, we found that its performance was suboptimal. Consequently, the model is primarily designed as a two-stage process, which will be discussed in detail in later sections. The overall structure consists of four phases: the training phase, zero-shot testing, one-stage image generation, and two-stage image generation, each contributing to the transformation of raw EEG data into meaningful visual outputs.

## 3.2 TRAINING PHASE

In the initial training phase, both visual image $\in \mathbb{R}^{h \times w \times ch}$ and EEG data $\in \mathbb{R}^{e \times d}$ are processed in parallel to establish a shared embedding space, where $h$ is the height of the image, $w$ is the width of the image, $ch$ is the number of channels (e.g., RGB channels), $e$ is the number of electrodes (channels), and $d$ is the number of data points (time samples). Training set images are first passed through a pre-trained image encoder, which transforms the images into latent representations called image embeddings **I**. In this work, we use a pretrained Vision Transformer (ViT) Dosovitskiy et al. (2020) from CLIP model Radford et al. (2021a) as the image encoder, which outputs embeddings of size $\mathbb{R}^{1 \times 1024}$ for each image. Simultaneously, the EEG signals from the corresponding sessions are processed by a custom EEG encoder to produce EEG embeddings **E**. As for the EEG encoder, in this work, we extended several existing works like NICE Song et al. (2024), MUSE Chen & Wei (2024), Nervformer Chen & Wei (2024) and ATM Li et al. (2024) to enable EEG-to-image capabilities. Additionally, we proposed a new EEG encoder, NERV, which is specifically designed for noisy, multi-channel time series data like EEG, based on a multi-attention mechanism.

These embeddings are projected into a unified space via an MLP Projector, where they are trained using the InfoNCE loss. This contrastive learning loss function ensures that corresponding image and EEG embeddings are aligned in the latent space, enhancing the model's ability to understand and link neural patterns to visual stimuli. Standard contrastive learning employs the InfoNCE loss as defined

by Oord et al. (2018); He et al. (2020); Radford et al. (2021b):

$$\mathcal{L}_{InfoNCE} = -\mathbb{E}\left[\log \frac{\exp(S_{\mathbf{E},\mathbf{I}}/\tau)}{\sum_{k=1}^{N} \exp(S_{\mathbf{E},\mathbf{I}_k}/\tau)}\right] \tag{1}$$

where the $S_{\mathbf{E},\mathbf{I}}$ represents the similarity score between the EEG embeddings $\mathbf{E}$, and the paired image embeddings $\mathbf{I}$, and the $\tau$ is learned temperature parameter.

### 3.3 ZERO-SHOT TESTING PHASE

Once trained, the model enters the zero-shot testing phase. This phase focuses on evaluating the model's ability to generalize to unseen data. Here, the EEG signals and images from the test set are encoded using the pre-trained encoders, and their respective embeddings are projected through the MLP Projector. The testing groups are separated into multiple divisions—2-way, 4-way, 10-way, 50-way, 100-way and beyond—allowing for a structured comparison between the EEG and image embeddings. The final similarity scores between embeddings determine the model's classification accuracy, enabling the assessment of how well the model understands new EEG data without additional training.

### 3.4 ONE-STAGE IMAGE GENERATION

In the one-stage image generation process, the EEG embeddings from the testing set are directly used as inputs to reconstruct images. By incorporating the IP-Adapter Ye et al. (2023), which was originally designed to use images as prompts, due to its compact design, enhances image prompt flexibility within pre-trained text-to-image models. We adapt it in this work as a means to transform EEG embeddings into "feature prompts" for the image generation process. The conditioned embeddings are then processed by the Stable Diffusion XL-Turbo model Podell et al. (2023); Luo et al. (2024), a faster version of Stable Diffusion XL designed for rapid image synthesis, which reconstructs the final images based on the input EEG data. This method offers a streamlined approach to EEG-based image generation, relying on a single transformation stage to produce meaningful visual outputs from neural signals. The start of the EEG-conditioned diffusion phase is critical for generating images based on EEG data. This phase uses a classifier-free guidance method, which pairs CLIP embeddings and EEG embeddings $(\mathbf{I}, \mathbf{E})$. By applying advanced generative techniques, the diffusion process is adapted to use the EEG embedding $\mathbf{E}$ to model the distribution of the CLIP embeddings $p(\mathbf{I}|\mathbf{E})$. The CLIP embedding $\mathbf{I}$, generated during this stage, lays the foundation for the next phase of image generation. The architecture integrates a simplified U-Net model, represented as $\epsilon_{\text{prior}}(\mathbf{I}^t, t, \mathbf{E})$, where $\mathbf{I}^t$ is the noisy CLIP embedding at a specific diffusion step $t$.

The classifier-free guidance method helps refine the diffusion model (DM) using a specific EEG condition $\mathbf{E}$. This approach synchronizes the outputs of both a conditional and an unconditional model. The final model equation is expressed as:

$$\epsilon_{\text{prior}}^{w}(\mathbf{I}^t, t, \mathbf{E}) = (1+w)\epsilon_{\text{prior}}(\mathbf{I}^t, t, \mathbf{E}) - w\epsilon_{\text{prior}}(\mathbf{I}^t, t), \tag{2}$$

where $w \geq 0$ controls the guidance scale. This technique allows for training both the conditional and unconditional models within the same network, periodically replacing the EEG embedding $\mathbf{E}$ with a null value to enhance training variation (about 10% of the data points). The main goal is to improve the quality of generated images while maintaining diversity.

However, we were surprised to find that when using EEG embeddings directly as prompts for the diffusion model, the generated images mostly turned out to be landscapes, regardless of the category. We will discuss the detailed results in later sections. As a result, we attempted a 2-stage approach for image generation.

### 3.5 TWO-STAGE IMAGE GENERATION

The prior diffusion stage plays a crucial role in generating an intermediate representation Zhu & Mumford (1997), such as a CLIP image embedding, from a text caption Ramesh et al. (2022). This representation is then used by the diffusion decoder to produce the final image. This two-stage

process enhances image diversity, maintains photorealism, and allows for efficient and controlled image generation Scotti et al. (2023). The two-stage image generation process introduces a more complex and refined method of synthesizing images from EEG data. In this approach, the EEG embeddings are first processed by a Diffusion U-Net, which applies additional transformations to enhance the representation of the neural data. After passing through the U-Net, the modified EEG embeddings are fed into the Stable Diffusion XL-Turbo model, with the assistance of the IP-Adaptor. This two-step transformation ensures a more nuanced generation process, potentially leading to higher-quality images by incorporating deeper layers of refinement. The first step of stage-1 is training the prior diffusion model. The main purpose of training is to let the model learn how to recover the original embedding from noisy embeddings. The specific steps are as follows: (a) Randomly replace conditional EEG embeddings $c_{\text{emb}}$ with None with a 10% probability:

$$c_{\text{emb}} = \text{None}, \quad \text{if random}() < 0.1 \tag{3}$$

(b) Add random noise to the target embedding $h_{\text{emb}}$, perturb it using the scheduler at a timestep $t$, use the symbol $\mathcal{S}_{add\_noise}$ to represent the scheduler add noise function:

$$\hat{h}_{\text{emb}}(t) = \mathcal{S}_{add\_noise}(h_{\text{emb}}, \epsilon, t) \tag{4}$$

where $\epsilon \sim \mathcal{N}(0, I)$ is the random noise, and $t$ is a randomly sampled timestep. (c) The model receives the perturbed embedding $\hat{h}_{\text{emb}}(t)$ and conditional embedding $c_{\text{emb}}$, and predicts the noise. Use the symbol $\mathcal{D}_{\text{prior}}$ to represent the diffusion prior function:

$$\epsilon_{\text{pred}} = \mathcal{D}_{\text{prior}}(\hat{h}_{\text{emb}}(t), t, c_{\text{emb}}) \tag{5}$$

(d) Compute the loss using Mean Squared Error (MSE) between the predicted noise and the actual noise:

$$L = \frac{1}{N} \sum_{i=1}^{N} \left( \epsilon_{\text{pred}}^{(i)} - \epsilon^{(i)} \right)^2 \tag{6}$$

(e) Perform backpropagation on the loss $L$, and update the model parameters using the optimizer:

$$\theta \leftarrow \theta - \eta \nabla_\theta L \tag{7}$$

where $\eta$ is the learning rate and $\theta$ represents the model's parameters.

The last step of stage-1 is generation process. The main purpose of the generation process is to gradually denoise and generate the final embedding based on the conditional EEG embedding $c_{\text{emb}}$, starting from random noise. The specific steps are as follows: (a) Generate a sequence of timesteps $t$, which will be used for the denoising process, define $\mathcal{T} = \{t_1, t_2, \ldots, t_T\}$ to represent the set of time steps sampled from the total steps $T$:

$$\{t_1, t_2, \ldots, t_T\} \sim \mathcal{T}(T) \tag{8}$$

where $T$ is the total number of denoising steps. (b) Initialize random noise embedding $h_T$, which serves as the starting point for the generation process:

$$h_T \sim \mathcal{N}(0, I) \tag{9}$$

(c) Starting from timestep $T$, iteratively apply the model to predict noise and denoise the embedding until $t = 0$. Each step depends on the conditional embedding $c_{\text{emb}}$:

If using conditional embedding, perform both unconditional and conditional noise prediction at each step:

$$\epsilon_{\text{pred\_cond}} = \mathcal{D}_{\text{prior}}(h_t, t, c_{\text{emb}}) \tag{10}$$
$$\epsilon_{\text{pred\_uncond}} = \mathcal{D}_{\text{prior}}(h_t, t) \tag{11}$$

Then combine the results using classifier-free guidance, define $\alpha_{\text{guide}}$ as the guidance scale:

$$\epsilon_{\text{pred}} = \epsilon_{\text{pred\_uncond}} + \alpha_{\text{guide}} \times (\epsilon_{\text{pred\_cond}} - \epsilon_{\text{pred\_uncond}}) \tag{12}$$

Finally, update the noisy embedding based on the predicted noise, use the symbol $\mathcal{S}_{step}$ to represent the scheduler step function:

$$h_{t-1} = \mathcal{S}_{step}(\epsilon_{\text{pred}}, t, h_t) \tag{13}$$

(d) After the denoising process is complete, $h_{output}$ represents the final generated embedding of a EEG, which is the model's output:

$$h_{output} = h_{\text{generated}} \in \mathbb{R}^{1 \times 1024} \tag{14}$$

The stage-2 is input the $h_{output}$ into the IP-adaptor as a prompt to generate the image by Stable Diffusion XL-Turbo model.

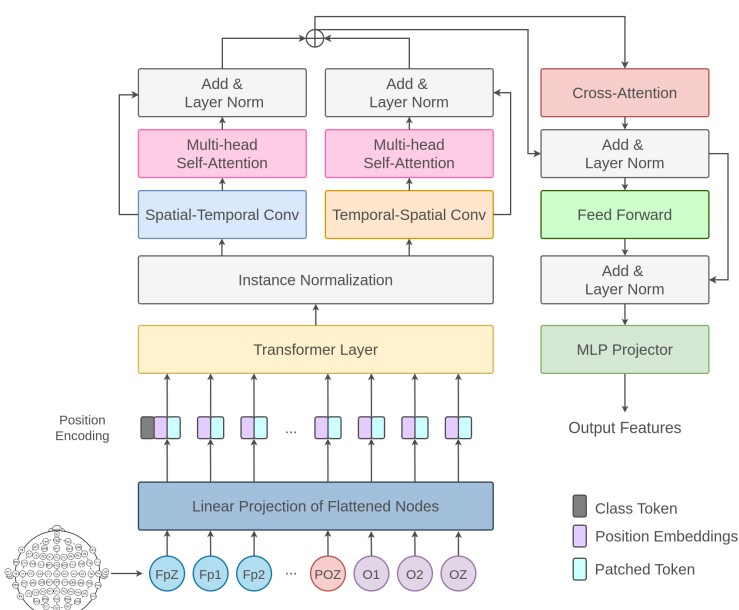

Figure 3: This diagram shows the overall structure and workflow of the NERV EEG encoder model.

## 3.6 NERV EEG ENCODER

This diagram 3 illustrates the structure of NERV, a neural network encoder designed for EEG signal processing. The workflow starts with a linear projection of the flattened EEG nodes, followed by position encoding to retain temporal information. EEG signals pass through a Transformer layer and undergo instance normalization. The model then applies both spatial-temporal convolution (blue) to extract spatial features followed by temporal features and temporal-spatial convolution (yellow) to extract temporal features first, then spatial features. Multi-head self-attention mechanisms are applied to both feature sets, followed by layer normalization and residual connections. The cross-attention block (red) fuses the temporal and spatial features, which are further processed by a feed-forward layer before final output. The class token, position embeddings, and patch tokens are all part of the input sequence processed through these steps, ultimately yielding the output features for EEG-based tasks.

## 3.7 CATEGORY-BASED ASSESSMENT TABLE (CAT) SCORE

Unlike traditional image-to-image or text-to-image models driven by image representations, EEG-to-image models face unique challenges. In the current NECOMIMI architecture, the model can only capture broad semantic information from EEG signals rather than fine-grained details. For example, suppose the ground truth EEG data was recorded while a subject was observing an aircraft carrier. When using Model A as the EEG encoder in NECOMIMI, the generated image is a jet, while using Model B results in an image of a sheep. To objectively assess performance, we need a standard that scores Model A higher than Model B in such cases.

Why not use existing evaluation metrics? Traditional metrics like Structural Similarity Index (SSIM) Wang et al. (2004) measure structural similarity between the ground truth and generated image, while the Inception Score (IS) Salimans et al. (2016) and Fréchet Inception Distance (FID) Heusel et al. (2017) focus on the accuracy of image categories and its distribution. However, EEG captures more abstract semantic information, and we cannot guarantee that the subject's thoughts during EEG recording perfectly align with the ground truth image. This makes traditional evaluation methods unfair for EEG-to-image tasks.

To address this, we propose the Category-Based Assessment Table (CAT) Score, a new metric specifically designed for EEG-to-image evaluation. In the ThingsEEG test dataset (which contains 200 categories with one image per category), each image is manually labeled with two tags for broad

categories, one for a specific category, and one for background content, resulting in a total of five tags per image. We extracted the tags by ChatGPT-4o OpenAI et al. (2023). The entire test dataset thus comprises 200 images × 5 tags = 1,000 points. Using manual annotation, we can determine whether the categories of generated images match these labels, providing a fair assessment for EEG-to-image models. For more details on the ThingsEEG categories, please refer to the appendix.

# 4 Experiments

## 4.1 Datasets and Preprocessing

The ThingsEEG dataset Gifford et al. (2022) consists of a large set of EEG recordings obtained through a rapid serial visual presentation (RSVP) paradigm. The responses were collected from 10 participants who viewed a total of 16,740 natural images from the THINGS database Hebart et al. (2019). The dataset contains 1654 training categories, each with 10 images, and 200 test categories, each with a single image. The EEG data were recorded using 64-channel EASYCAP equipment, and preprocessing involved segmenting the data into trials from 0 to 1000 ms after the stimulus was shown, with baseline correction based on the pre-stimulus period. EEG responses for each image were averaged over multiple repetitions.

## 4.2 Experiment Details

Due to the significant impact that different versions of the CLIP package can have on the results of contrastive learning, this work ensures a fair comparison of various EEG encoders by rerunning all experiments using a unified CLIP-ViT environment, where available open-source code (e.g., Song et al. (2024)[1], Chen & Wei (2024)[2], Li et al. (2024)[3]) was utilized. Another factor that can influence contrastive learning is batch size. Therefore, all experiments in this work were conducted with a batch size of 1024. The final results are averaged from the best outcomes of 5 random seed training sessions, each running for 200 epochs. We employ the AdamW optimizer, setting the learning rate to 0.0002 and parameters $\beta_1$=0.5 and $\beta_2$=0.999. The $\tau$ in contrastive learning initialized with $log(1/0.07)$. The NERV model achieves the best results with 5 multi-heads, while the Transformer layer has 1 multi-head and the cross-attention layer has 8 multi-heads. The time step is 50 in diffusion model. All experiments, including both EEG encoder training and prior diffusion model processing, were performed on a machine equipped with an A100 GPU.

## 4.3 Classification Results

In Table 1, the classification accuracy for both 2-way and 4-way zero-shot tasks is evaluated across ten subjects. Our new model NERV consistently achieves the best performance, particularly excelling in the 2-way classification task, where it maintains top accuracy across most subjects. It achieves an average accuracy of 94.8% in the 2-way classification and 86.8% in the 4-way classification, outperforming other methods like NICE Song et al. (2024), MUSE Chen & Wei (2024), and ATM-S Li et al. (2024). While NICE and MUSE perform strongly in some subjects, they often fall short of NERV's performance. NICE has an average of 91.3% in the 2-way task and 81.3% in the 4-way task, with MUSE trailing behind with averages of 92.2% (2-way) and 82.8% (4-way). ATM-S performs comparably to NICE and MUSE in some subjects but falls short on average with 86.5% in the 4-way classification. In Table 2, the results for the more challenging 200-way zero-shot classification task show that NERV also performs the best, especially in the top-1 accuracy. ATM-S and NERV perform similarly, but NERV shows stronger performance in most subjects. NERV achieves an average top-1 accuracy of 27.9% and top-5 accuracy of 54.7%, leading over all other methods. In contrast, Nervformer Chen & Wei (2024) and BraVL Du et al. (2023) show weaker performance, especially in the top-1 accuracy, where they average 19.8% and 5.8%, respectively. For the results of other 10-way, 50-way, and 100-way zero-shot classifications, please refer to the appendix. In summary, NERV consistently outperforms its competitors in both tasks, demonstrating the strongest zero-shot

---

[1] https://github.com/eeyhsong/NICE-EEG
[2] https://github.com/ChiShengChen/MUSE_EEG
[3] https://github.com/dongyangli-del/EEG_Image_decode

classification capability, particularly when distinguishing between a large number of categories, making it the most effective model in these experiments.

Table 1: Overall accuracy (%) of 2-way and 4-way zero-shot classification using CLIP-ViT as image encoder: top-1 and top-5. The parts in bold represent the best results, while the underlined parts are the second best.

| Method | Subject 1 | | Subject 2 | | Subject 3 | | Subject 4 | | Subject 5 | | Subject 6 | | Subject 7 | | Subject 8 | | Subject 9 | | Subject 10 | | Ave | |
|---|---|---|---|---|---|---|---|---|---|---|---|---|---|---|---|---|---|---|---|---|---|---|
| | 2-way | 4-way | 2-way | 4-way | 2-way | 4-way | 2-way | 4-way | 2-way | 4-way | 2-way | 4-way | 2-way | 4-way | 2-way | 4-way | 2-way | 4-way | 2-way | 4-way | 2-way | 4-way |
| Subject dependent - train and test on one subject | | | | | | | | | | | | | | | | | | | | | | |
| Nervformer | 89.9 | 76.9 | 91.3 | 80.7 | 91.6 | 80.8 | 94.3 | 85.9 | 86.3 | 70.4 | 91.1 | 82.5 | 92.5 | 81.6 | 96.2 | 88.3 | 92.0 | 83.7 | 92.4 | 83.1 | 91.8 | 81.4 |
| NICE | 91.7 | 80.4 | 89.8 | 77.4 | 93.5 | 83.7 | 94.0 | 84.9 | 85.9 | 70.3 | 89.1 | 81.7 | 91.2 | 81.7 | 95.8 | 89.2 | 87.9 | 76.5 | 93.8 | 87.1 | 91.3 | 81.3 |
| MUSE | 90.1 | 78.4 | 90.3 | 76.8 | 93.4 | 85.6 | 93.6 | 87.5 | 88.3 | 74.2 | 93.1 | 85.3 | 93.1 | 82.8 | 95.4 | 87.7 | 90.5 | 81.8 | 94.4 | 88.1 | 92.2 | 82.8 |
| ATM-S | 94.8 | 84.9 | 93.5 | 86.3 | 95.3 | 89.0 | 95.9 | 87.3 | 90.8 | 78.5 | 94.1 | 85.2 | 94.2 | 87.1 | 96.6 | 92.9 | 94.1 | 86.8 | 94.7 | 87.0 | 94.4 | 86.5 |
| NERV (ours) | 95.3 | 85.7 | 96.0 | 88.8 | 95.9 | 91.2 | 95.8 | 87.4 | 90.8 | 80.4 | 93.6 | 84.0 | 94.7 | 86.2 | 96.8 | 92.3 | 94.4 | 84.2 | 94.8 | 87.6 | **94.8** | **86.8** |

Table 2: Overall accuracy (%) of 200-way zero-shot classification using CLIP-ViT as image encoder: top-1 and top-5. The parts in bold represent the best results, while the underlined parts are the second best.

| Method | Subject 1 | | Subject 2 | | Subject 3 | | Subject 4 | | Subject 5 | | Subject 6 | | Subject 7 | | Subject 8 | | Subject 9 | | Subject 10 | | Ave | |
|---|---|---|---|---|---|---|---|---|---|---|---|---|---|---|---|---|---|---|---|---|---|---|
| | top-1 | top-5 | top-1 | top-5 | top-1 | top-5 | top-1 | top-5 | top-1 | top-5 | top-1 | top-5 | top-1 | top-5 | top-1 | top-5 | top-1 | top-5 | top-1 | top-5 | top-1 | top-5 |
| Subject dependent - train and test on one subject | | | | | | | | | | | | | | | | | | | | | | |
| BraVL | 6.1 | 17.9 | 4.9 | 14.9 | 5.6 | 17.4 | 5.0 | 15.1 | 4.0 | 13.4 | 6.0 | 18.2 | 6.5 | 20.4 | 8.8 | 23.7 | 4.3 | 14.0 | 7.0 | 19.7 | 5.8 | 17.5 |
| Nervformer | 15.0 | 36.7 | 15.6 | 40.0 | 19.7 | 44.9 | 23.3 | 54.4 | 13.0 | 29.1 | 18.9 | 42.2 | 19.5 | 42.0 | 30.3 | 60.0 | 20.1 | 46.3 | 22.9 | 47.1 | 19.8 | 44.3 |
| NICE | 19.3 | 44.8 | 15.2 | 38.2 | 23.9 | 51.4 | 24.1 | 51.6 | 11.0 | 30.7 | 18.5 | 43.8 | 21.0 | 47.9 | 32.5 | 63.5 | 18.2 | 42.4 | 27.4 | 57.1 | 21.1 | 47.1 |
| MUSE | 19.8 | 41.1 | 15.3 | 34.2 | 24.7 | 52.6 | 24.7 | 52.6 | 12.1 | 33.7 | 22.1 | 51.9 | 21.0 | 48.6 | 33.2 | 59.9 | 19.1 | 43.0 | 25.0 | 55.2 | 21.7 | 47.3 |
| ATM-S | 25.8 | 54.1 | 24.6 | 52.6 | 28.4 | 62.9 | 25.9 | 57.8 | 16.2 | 41.9 | 21.2 | 53.0 | 25.9 | 57.2 | 37.9 | 71.1 | 26.0 | 53.9 | 30.0 | 60.9 | 26.2 | **56.5** |
| NERV (ours) | 25.4 | 51.2 | 24.1 | 51.1 | 28.6 | 53.9 | 30.0 | 58.4 | 19.3 | 43.9 | 24.9 | 52.3 | 26.1 | 51.6 | 40.8 | 67.4 | 27.0 | 55.2 | 32.3 | 61.6 | **27.9** | 54.7 |

## 4.4 Performance Comparison of different generative models

Here, we introduce our newly proposed CAT Score method, which quantifies and evaluates the quality of EEG-generated images based on semantic concepts rather than pixel structure. Detailed CAT Score labels can be found in the appendix. To our surprise, while our proposed NERV method achieved SoTA on the CAT Score, no EEG encoder has surpassed a score of 500 in this evaluation out of a possible 1000 points. This highlights both the rigor of the CAT Score and the challenging nature of the pure EEG-to-Image task.

Table 3: Overall CAT score ×1000 of NECOMIMI EEG-to-Image generation with several EEG encoders.

| EEG Encoder | Subject 1 | Subject 2 | Subject 3 | Subject 4 | Subject 5 | Subject 6 | Subject 7 | Subject 8 | Subject 9 | Subject 10 | Ave |
|---|---|---|---|---|---|---|---|---|---|---|---|
| | | | | | | CAT Score | | | | | |
| Nervformer | 432 | 457 | 429 | 454 | 475 | 463 | 404 | 438 | 427 | 410 | 438.9 |
| NICE | 426 | 456 | 445 | 447 | 411 | 454 | 438 | 443 | 426 | 429 | 437.5 |
| MUSE | 438 | 456 | 434 | 416 | 426 | 463 | 443 | 437 | 410 | 468 | 439.1 |
| ATM-S | 413 | 419 | 411 | 464 | 427 | 469 | 442 | 472 | 431 | 445 | 439.3 |
| NERV (ours) | 445 | 436 | 432 | 456 | 438 | 466 | 410 | 437 | 433 | 444 | **439.7** |

## 4.5 Findings in EEG-to-Image

We have observed some interesting findings from the pure EEG-to-Image process. As shown in the third row of Figure 4, the images generated by the diffusion model from embeddings compressed from EEG signals mainly consist of landscapes, which differ significantly from the original images (ground truth). Several factors may contribute to this phenomenon. For example, EEG signals are a high-noise, low-resolution form of data, capturing only certain aspects of brain activity. Moreover, we are currently unable to assess whether the brainwave data recorded from the subjects accurately captures the complete information of the original images, as the subjects might have been distracted and thinking about other things during the recording. This makes it difficult for the embeddings extracted from EEG signals to capture sufficient details, particularly when it comes to high-resolution object recognition (such as cats or specific items). As a result, the model tends to generate relatively vague or abstract images, like landscapes. Alternatively, the EEG signals may reflect higher-level abstract concepts or emotions associated with viewing the images rather than concrete objects or

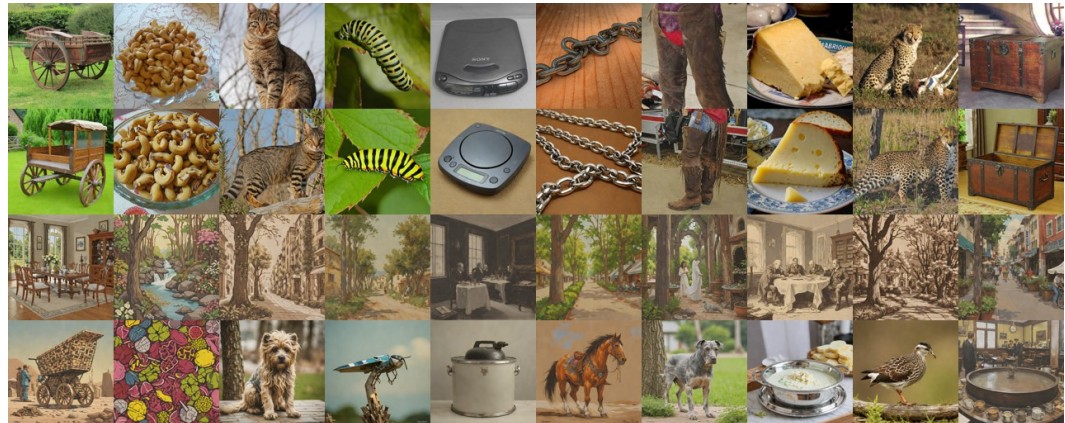

Figure 4: The image illustrates the progression of visual representations generated using different embedding techniques in a diffusion model: (a) Top row: The original images shown to subjects (ground truth). (b) Second row: Images generated by the CLIP-ViT embeddings of the original images. (c) Third row: Images generated by one-stage method using pure EEG embeddings with NERV EEG encoder. (d) Fourth row: Images generated by two-stage NECOMIMI method using pure EEG embeddings with NERV EEG encoder.

details. Since these abstract concepts are often related to the scene, background, or the brain's broad perception of the environment, the model is more likely to generate abstract or general images, such as landscapes, instead of specific objects.

Additionally, the training of the model on EEG signals may still be insufficient. The diffusion model may not yet fully understand and generate images from EEG signals, especially when it lacks enough data or optimization to map EEG signals to specific visual information. As a result, the model might more easily generate the types of images it is "accustomed" to producing, such as landscapes, which may constitute a significant portion of the training data. The gap between the vision modality and the neural modality (EEG) is also substantial. EEG signals may not directly correspond to detailed objects in images, so the model tends to generate "safe options," like landscapes, which may have been more prevalent in the image generation samples during training. This leads to what can be described as "hallucinations." These factors collectively contribute to the significant differences between the images generated from EEG signals and the ground truth, particularly the failure in specific object recognition. This work can be considered a forward-looking exploration, as this field is just beginning to develop.

## 5 DISCUSSION AND CONCLUSION

The NECOMIMI framework expands previous works on EEG-Image contrastive learning classification by enabling image generation, filling a gap in prior research and opening new possibilities for EEG applications. We introduced the SoTA EEG encoder, NERV, which achieved top performance in 2-way, 4-way, and 200-way zero-shot classification tasks, as well as in the CAT Score evaluation, demonstrating its effectiveness in EEG-based generative tasks. A key finding is that the model often generates abstract images, like landscapes, rather than specific objects. This suggests that EEG data, being noisy and low-resolution, captures broad semantic concepts rather than detailed visuals. The gap between neural signals and visual stimuli remains a challenge for precise image generation. We also proposed the CAT Score, a new metric tailored for EEG-to-image generation, and established its benchmark on the ThingsEEG dataset. Surprisingly, we found that EEG encoder performance may not strongly correlate with the quality of generated images, providing new insights into the limitations and challenges of this task. In conclusion, NECOMIMI demonstrates the potential of EEG-to-image generation while highlighting the complexities of translating neural signals into accurate visual representations. Future research should focus on refining models to better capture detailed information from EEG signals.

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

# A APPENDIX

## A.1 MORE EEG ENCODER CLASSIFICATION PERFORMANCE COMPARISON

Table 4: Overall accuracy (%) of 10-way zero-shot classification using CLIP-ViT as image encoder: top-1 and top-5.

| | Subject 1 | Subject 2 | Subject 3 | Subject 4 | Subject 5 | Subject 6 | Subject 7 | Subject 8 | Subject 9 | Subject 10 | Ave |
|---|---|---|---|---|---|---|---|---|---|---|---|
| Method | 10-way | 10-way | 10-way | 10-way | 10-way | 10-way | 10-way | 10-way | 10-way | 10-way | 10-way |
| Subject dependent - train and test on one subject | | | | | | | | | | | |
| Nervformer | 59.4 | 62.0 | 65.4 | 72.0 | 50.7 | 63.4 | 63.7 | 78.3 | 67.0 | 68.8 | 65.1 |
| NICE | 64.1 | 57.6 | 70.2 | 72.6 | 51.8 | 63.0 | 63.8 | 79.1 | 59.6 | 73.9 | 65.6 |
| MUSE | 61.0 | 56.1 | 70.8 | 71.3 | 55.1 | 70.1 | 66.2 | 76.9 | 62.8 | 73.2 | 66.4 |
| ATM-S | 72.5 | 70.4 | 76.3 | 74.1 | 64.6 | 72.2 | 73.6 | 83.2 | 70.6 | 75.8 | 73.3 |
| NERV (ours) | 72.2 | 74.3 | 75.9 | 76.7 | 62.5 | 71.8 | 70.4 | 81.8 | 70.9 | 73.8 | 73.0 |

Table 5: Overall accuracy (%) of 50-way zero-shot classification using CLIP-ViT as image encoder: top-1 and top-5.

| | Subject 1 | | Subject 2 | | Subject 3 | | Subject 4 | | Subject 5 | | Subject 6 | | Subject 7 | | Subject 8 | | Subject 9 | | Subject 10 | | Ave | |
|---|---|---|---|---|---|---|---|---|---|---|---|---|---|---|---|---|---|---|---|---|---|---|
| Method | top-1 | top-5 | top-1 | top-5 | top-1 | top-5 | top-1 | top-5 | top-1 | top-5 | top-1 | top-5 | top-1 | top-5 | top-1 | top-5 | top-1 | top-5 | top-1 | top-5 | top-1 | top-5 |
| Subject dependent - train and test on one subject | | | | | | | | | | | | | | | | | | | | | | |
| Nervformer | 28.4 | 66.0 | 32.0 | 71.8 | 37.4 | 73.9 | 44.8 | 81.6 | 24.6 | 57.1 | 33.8 | 74.4 | 33.6 | 69.2 | 49.9 | 87.2 | 36.8 | 75.6 | 38.8 | 76.6 | 36.0 | 73.3 |
| NICE | 36.0 | 72.2 | 30.2 | 66.8 | 43.0 | 77.8 | 44.0 | 80.3 | 24.8 | 58.2 | 35.6 | 70.4 | 36.9 | 72.5 | 53.3 | 86.0 | 34.4 | 65.4 | 45.8 | 82.8 | 38.4 | 73.2 |
| MUSE | 33.9 | 70.9 | 29.9 | 65.7 | 43.6 | 79.4 | 42.8 | 79.8 | 26.1 | 63.4 | 39.8 | 79.4 | 39.8 | 73.3 | 49.8 | 84.2 | 34.4 | 72.7 | 44.5 | 81.1 | 38.5 | 74.9 |
| ATM-S | 45.3 | 78.7 | 44.5 | 80.5 | 49.8 | 85.0 | 46.2 | 83.2 | 33.3 | 69.2 | 42.8 | 81.1 | 47.5 | 80.8 | 59.7 | 91.0 | 45.8 | 79.3 | 50.6 | 82.4 | 46.6 | 81.1 |
| NERV (ours) | 41.1 | 74.8 | 43.2 | 80.5 | 47.9 | 82.8 | 48.1 | 83.5 | 36.4 | 70.7 | 43.0 | 77.6 | 43.5 | 77.3 | 59.2 | 88.4 | 46.1 | 79.4 | 51.0 | 81.7 | 46.0 | 79.7 |

Table 6: Overall accuracy (%) of 100-way zero-shot classification using CLIP-ViT as image encoder: top-1 and top-5.

| | Subject 1 | | Subject 2 | | Subject 3 | | Subject 4 | | Subject 5 | | Subject 6 | | Subject 7 | | Subject 8 | | Subject 9 | | Subject 10 | | Ave | |
|---|---|---|---|---|---|---|---|---|---|---|---|---|---|---|---|---|---|---|---|---|---|---|
| Method | top-1 | top-5 | top-1 | top-5 | top-1 | top-5 | top-1 | top-5 | top-1 | top-5 | top-1 | top-5 | top-1 | top-5 | top-1 | top-5 | top-1 | top-5 | top-1 | top-5 | top-1 | top-5 |
| Subject dependent - train and test on one subject | | | | | | | | | | | | | | | | | | | | | | |
| Nervformer | 21.0 | 50.8 | 21.6 | 55.1 | 27.6 | 58.5 | 33.0 | 67.8 | 17.0 | 43.4 | 24.7 | 56.2 | 24.5 | 54.8 | 39.8 | 75.6 | 26.8 | 62.3 | 30.2 | 63.6 | 26.6 | 58.8 |
| NICE | 28.0 | 60.5 | 21.8 | 53.2 | 33.1 | 64.2 | 32.2 | 65.9 | 16.8 | 43.9 | 26.0 | 57.6 | 28.0 | 59.0 | 40.7 | 76.0 | 24.5 | 54.5 | 37.2 | 71.0 | 28.8 | 60.6 |
| MUSE | 25.4 | 56.7 | 21.2 | 49.8 | 33.9 | 67.6 | 32.2 | 65.7 | 18.0 | 49.6 | 30.4 | 67.2 | 29.5 | 60.8 | 39.0 | 73.3 | 26.1 | 58.7 | 33.6 | 67.0 | 28.9 | 61.6 |
| ATM-S | 34.9 | 67.7 | 33.1 | 66.9 | 38.1 | 74.3 | 36.0 | 70.2 | 24.6 | 55.6 | 28.4 | 67.4 | 35.1 | 67.9 | 48.3 | 82.1 | 33.2 | 68.6 | 39.1 | 73.0 | 35.1 | 69.4 |
| NERV (ours) | 31.1 | 64.4 | 33.1 | 66.9 | 36.6 | 74.1 | 39.0 | 70.2 | 26.1 | 57.1 | 32.9 | 65.2 | 34.2 | 66.0 | 50.4 | 78.0 | 35.5 | 67.7 | 41.1 | 72.5 | 36.0 | 68.2 |

## A.2 DETAILS OF CATEGORY-BASED ASSESSMENT TABLE (CAT) SCORE

All the category-based labels are generated by ChatGPT-4o [4], the prompt we used is "Please provide me with 5 one-word descriptions of the image, ranging from high level to low level.".

| Image Label | Test Image in ThingsEEG | Category-based label |
|---|---|---|
| 00001_aircraft_carrier | | Ship   Carrier   Deck   Island   Carrier   Antenna |
| | | *Continued on next page* |

---

[4] https://chatgpt.com

| Image Label | Test Image in ThingsEEG | Category-based label |
|---|---|---|
| 00002_antelope | | Animal   Antelope   Fur Grassland   Horns |
| 00003_backscratcher | | Object   Tool   Backscratcher Wood   Handle |
| 00004_balance_beam | | Structure   Beam   Wood Grass   Support |
| 00005_banana | | Fruit   Banana   Yellow Spotted   Plate |
| 00006_baseball_bat | | Sports   Bats   Baseball Black   Grass |
| 00007_basil | | Plant   Herb   Basil Green   Leaves |
| 00008_basketball | | Sport   Basketball   Ball Orange   Court |
| | | *Continued on next page* |

| Image Label | Test Image in ThingsEEG | Category-based label | | |
|---|---|---|---|---|
| 00009_bassoon |  | Instrument Stage | Bassoon Chair | Woodwind |
| 00010_baton4 |  | Race Yellow | Relay Hand | Baton |
| 00011_batter |  | Cooking Whisk | Batter Bowl | Mixing |
| 00012_beaver |  | Animal Tail | Beaver Paws | Fur |
| 00013_bench |  | Outdoor Garden | Bench Trees | Wooden |
| 00014_bike |  | Bicycle Frame | Road Path | Wheels |
| 00015_birthday_cake |  | Cake Pink | Candles Frosting | Flames |

| Image Label | Test Image in ThingsEEG | Category-based label |
|---|---|---|
| 00016_blowtorch |  | Tool     Blowtorch     Flame Canister     Gas |
| 00017_boat |  | Boat     Water     Blue Old     Rowing |
| 00018_bok_choy |  | Vegetable     BokChoy     Green Leafy     Stems |
| 00019_bonnet |  | Hat     Bonnet     Ribbon Fabric     Vintage |
| 00020_bottle_opener |  | Tool     Opener     Wooden Bottlecap     Engraving |
| 00021_brace |  | Support     Brace     Joint Black     Strap |
| 00022_bread |  | Food     Bread     Loaf Slice     Crust |

| Image Label | Test Image in ThingsEEG | Category-based label |
|---|---|---|
| 00023_breadbox |  | Storage    Breadbox    Wooden Bread    Countertop |
| 00024_bug |  | Insect    Bug    Leaf Brown    Antennae |
| 00025_buggy |  | Vehicle    Buggy    Off-road Wheels    Helmet |
| 00026_bullet |  | Ammunition    Bullet    Brass Cartridge    Metal |
| 00027_bun |  | Food    Bun    Sesame Bread    Round |
| 00028_bush |  | Plants    Bushes    Green Mulch    Shrub |
| 00029_calamari |  | Food    Calamari    Fried Plate    Lemon |

| Image Label | Test Image in ThingsEEG | Category-based label |
|---|---|---|
| 00030_candlestick |  | Candlesticks   Brass   Holders   Antique   Table |
| 00031_cart |  | Cart   Wheels   Wooden   Farm   Grass |
| 00032_cashew |  | Nuts   Cashews   Bowl   Snack   Glass |
| 00033_cat |  | Animal   Cat   Tabby   Fur   Whiskers |
| 00034_caterpillar |  | Insect   Caterpillar   Striped   Green   Leaf |
| 00035_cd_player |  | Device   CDPlayer   Portable   Gray   Buttons |
| 00036_chain |  | Metal   Chain   Links   Rusty   Wood |
| | | *Continued on next page* |

| Image Label | Test Image in ThingsEEG | Category-based label |
|---|---|---|
| 00037_chaps | | Clothing    Chaps    Leather
Fringe    Brown |
| 00038_cheese | | Food    Cheese    Wedge
Yellow    Cracker |
| 00039_cheetah | | Animal    Cheetah    Spotted
Hunt    Grassland |
| 00040_chest2 | | Furniture    Chest    Wooden
Vintage    Lock |
| 00041_chime | | Instrument    Chime    Percussion
Metal    Stand |
| 00042_chopsticks | | Utensils    Chopsticks    Wooden
Metal    Case |
| 00043_cleat | | Footwear    Cleats    Shoe
Green    Studs |

*Continued on next page*

| Image Label | Test Image in ThingsEEG | Category-based label |
|---|---|---|
| 00044_cleaver |  | Tool    Cleaver    Blade
Handle    Steel |
| 00045_coat |  | Clothing    Coat    Black
Double-breasted    Hanger |
| 00046_cobra |  | Animal    Cobra    Snake
Hood    Sand |
| 00047_coconut |  | Fruit    Coconut    Shell
White    Husk |
| 00048_coffee_bean |  | Coffee    Beans    Roasted
Brown    Grinder |
| 00049_coffeemaker |  | Appliance    Coffeemaker    Machine
Carafe    Buttons |
| 00050_cookie |  | Cookies    Snack    Chocolate
Stack    Crumb |

*Continued on next page*

| Image Label | Test Image in ThingsEEG | Category-based label |
|---|---|---|
| 00051_cordon_bleu |  | Food    Chicken    CordonBleu Breaded    Stuffed |
| 00052_coverall |  | Clothing    Coverall    Workwear Pockets    Green |
| 00053_crab |  | Animal    Crab    Beach Claws    Sand |
| 00054_creme_brulee |  | Dessert    CrèmeBrûlée    Caramelized Custard    Spoon |
| 00055_crepe |  | Dessert    Crepe    Chocolate Banana    Plate |
| 00056_crib |  | Furniture    Crib    Wooden Baby    Bedding |
| 00057_croissant |  | Pastry    Croissant    Flaky Golden    Plate |

| Image Label | Test Image in ThingsEEG | Category-based label |
|---|---|---|
| 00058_crow |  | Bird   Crow   Black
Feathers   Beak |
| 00059_cruise_ship |  | Vessel   Cruise   Ship
Ocean   Deck |
| 00060_crumb |  | Crumbs   Plate   Food
Leftovers   White |
| 00061_cupcake |  | Cupcake   Dessert   Chocolate
Icing   Wrapper |
| 00062_dagger |  | Weapon   Dagger   Blade
Handle   Rock |
| 00063_dalmatian |  | Dog   Dalmatian   Spotted
White   Grass |
| 00064_dessert |  | Dessert   Berries   Cream
Trifle   Glass |

*Continued on next page*

| Image Label | Test Image in ThingsEEG | Category-based label | | |
|---|---|---|---|---|
| 00065_dragonfly |  | Insect Striped | Dragonfly Branch | Wings |
| 00066_dreidel |  | Toy Spinning | Dreidel Letters | Wooden |
| 00067_drum |  | Instrument Blue | Drum Percussion | Sticks |
| 00068_duffel_bag |  | Bag Straps | Container Eagles | Green |
| 00069_eagle |  | Bird Wings | Eagle Sky | Flight |
| 00070_eel |  | Fish Tank | Eel Gravel | Aquatic |
| 00071_egg |  | Eggs Food | Bowl Shell | Brown |
| | | *Continued on next page* | | |

| Image Label | Test Image in ThingsEEG | Category-based label | | |
|---|---|---|---|---|
| 00072_elephant |  | Animal Zoo | Elephant Mammal | Trunk |
| 00073_espresso |  | Drink Coffee | Espresso Saucer | Cup |
| 00074_face_mask |  | Gear Cage | Mask Protection | Helmet |
| 00075_ferry |  | Ferry Water | Boat Orange | Transport |
| 00076_flamingo |  | Bird Water | Flamingo Beach | Pink |
| 00077_folder |  | Folder Papers | Office Desk | Orange |
| 00078_fork |  | Utensil Plate | Fork Tablecloth | Silver |
| | | *Continued on next page* | | |

| Image Label | Test Image in ThingsEEG | Category-based label |
|---|---|---|
| 00079_freezer |  | Appliance Freezer Storage
Cold White |
| 00080_french_horn |  | Instrument Horn Brass
Coiled Shiny |
| 00081_fruit |  | Fruits Assortment Tropical
Colorful Fresh |
| 00082_garlic |  | Garlic Bulb Cloves
White Peeled |
| 00083_glove |  | Gloves Knitted Patterned
Wool Gray |
| 00084_golf_cart |  | Vehicle GolfCart White
Seats Wheels |
| 00085_gondola |  | Boats Gondolas Venice
Water Blue |
| | | *Continued on next page* |

| Image Label | Test Image in ThingsEEG | Category-based label |
|---|---|---|
| 00086_goose |  | Bird   Goose   Flight
Wings   Lake |
| 00087_gopher |  | Animal   Gopher   Furry
Rodent   Field |
| 00088_gorilla |  | Animal   Gorilla   Primates
Silverback   Grass |
| 00089_grasshopper |  | Insect   Grasshopper   Antennae
Legs   Green |
| 00090_grenade |  | Weapon   Grenade   Metal
Pin   Explosive |
| 00091_hamburger |  | Food   Hamburger   Bun
Lettuce   Grilled |
| 00092_hammer |  | Tool   Hammer   Handle
Metal   Claw |

| Image Label | Test Image in ThingsEEG | Category-based label | | |
|---|---|---|---|---|
| 00093_handbrake |  | Automobile Lever | Interior Grip | Handbrake |
| 00094_headscarf |  | Headwear Pink | Scarf Wrap | Fabric |
| 00095_highchair |  | Red Highchair | Wooden Furniture | Chair |
| 00096_hoodie |  | White Casual | Hoodie Clothing | Ground |
| 00097_hummingbird |  | Hummingbird Small | Green Bird | Feeder |
| 00098_ice_cube |  | Ice Clear | Cold Cubes | Frozen |
| 00099_ice_pack |  | Gel Cold | Blue Cooling | Reusable |

| Image Label | Test Image in ThingsEEG | Category-based label |
|---|---|---|
| 00100_jeep |  | Off-road    Rugged    SUV
Adventure    Durable |
| 00101_jelly_bean |  | Colorful    Sweet    Candy
Vibrant    Chewy |
| 00102_jukebox |  | Retro    Vibrant    Music
Neon    Classic |
| 00103_kettle |  | Shiny    Stovetop    Whistling
Metallic    Classic |
| 00104_kneepad |  | Protective    Sporty    Durable
Cushioned    Ergonomic |
| 00105_ladle |  | Stainless    Sleek    Functional
Polished    Culinary |
| 00106_lamb |  | Adorable    Fluffy    Playful
Animal    Lamb |

| Image Label | Test Image in ThingsEEG | Category-based label | | |
|---|---|---|---|---|
| 00107_lampshade |  | Vintage Fringed | Floral Ornate | Fabric |
| 00108_laundry_basket |  | Laundry Towels | Plastic Grid | Basket |
| 00109_lettuce |  | Vegetable Fresh | Lettuce Green | Leafy |
| 00110_lightning_bug |  | Insect Glowing | Firefly Segmented | Antennae |
| 00111_manatee |  | Aquatic Mammal | Manatee Floating | Underwater |
| 00112_marijuana |  | Cannabis Leaves | Plant Green | Buds |
| 00113_meatloaf |  | Food Sauce | Meatloaf Hearty | Slice |

| Image Label | Test Image in ThingsEEG | Category-based label |
|---|---|---|
| 00114_metal_detector |  | Equipment    Detectors    Metal
Beach    Lineup |
| 00115_minivan |  | Vehicle    Minivan    Car
Blue    Electric |
| 00116_modem |  | Device    Modem    Router
Black    Connectivity |
| 00117_mosquito |  | Insect    Mosquito    Biting
Legs    Proboscis |
| 00118_muff |  | Accessory    Muff    Fur
Warm    Pink |
| 00119_music_box |  | Device    Music    Box
Crank    Punched |
| 00120_mussel |  | Seafood    Mussels    Shells
Steamed    Parsley |
| | | *Continued on next page* |

| Image Label | Test Image in ThingsEEG | Category-based label |
|---|---|---|
| 00121_nightstand |  | Furniture   Nightstand   Wooden   Drawer   Lamp |
| 00122_okra |  | Vegetable   Okra   Green   Basket   Fresh |
| 00123_omelet |  | Breakfast   Omelet   Vegetables   Tomatoes   Herbs |
| 00124_onion |  | Vegetable   Onion   Red   Sliced   Raw |
| 00125_orange |  | Fruit   Orange   Citrus   Sliced   Juicy |
| 00126_orchid |  | Flower   Orchid   Yellow   Bloom   Petals |
| 00127_ostrich |  | Bird   Ostrich   Large   Plumage   Road |

| Image Label | Test Image in ThingsEEG | Category-based label |
|---|---|---|
| 00128_pajamas |  | Clothing   Pajamas   Striped
Blue   Fabric |
| 00129_panther |  | Animal   Panther   Black
Predator   Stealthy |
| 00130_paperweight |  | Office   Paperwork   Paperweight
Eyeball   Documents |
| 00131_pear |  | Fruit   Pear   Tree
Green   Ripe |
| 00132_pepper1 |  | Spice   Pepper   Ground
Black   Spoon |
| 00133_pheasant |  | Bird   Pheasant   Feathers
Colorful   Wild |
| 00134_pickax |  | Tool   Pickaxe   Wooden
Metal   Digging |

| Image Label | Test Image in ThingsEEG | Category-based label |
|---|---|---|
| 00135_pie |  | Dessert Pie Baked Crust Golden |
| 00136_pigeon |  | Bird Pigeon Grey Perched Feathers |
| 00137_piglet |  | Animal Piglet Spotted Grass Cute |
| 00138_pocket |  | Clothing Jeans Pocket Denim Stitched |
| 00139_pocketknife |  | Tool Pocketknife Blade Compact Multi-functional |
| 00140_popcorn |  | Snack Popcorn Bowl Buttery Crispy |
| 00141_popsicle |  | Dessert Popsicle Colorful Frozen Fruit |
| | | *Continued on next page* |

| Image Label | Test Image in ThingsEEG | Category-based label |
|---|---|---|
| 00142_possum |  | Animal   Possum   Furry   Marsupial   Wild |
| 00143_pretzel |  | Snack   Pretzel   Salted   Baked   Dough |
| 00144_pug |  | Animal   Pug   Dog   Leash   Panting |
| 00145_punch2 |  | Tool   Punch   Metal   Office   Desk |
| 00146_purse |  | Accessory   Purse   Leather   Green   Handles |
| 00147_radish |  | Vegetable   Radish   Root   Fresh   Bunch |
| 00148_raspberry |  | Fruit   Raspberry   Red   Berry   Branch |

*Continued on next page*

| Image Label | Test Image in ThingsEEG | Category-based label |
|---|---|---|
| 00149_recorder |  | Instrument     Recorder     Music Notes     Sheet |
| 00150_rhinoceros |  | Animal     Rhinoceros     Horned Savanna     Wild |
| 00151_robot |  | Robot     Toy     Humanoid Black     White |
| 00152_rooster |  | Bird     Rooster     Feathers Colorful     Comb |
| 00153_rug |  | Furniture     Rug     Patterned Red     Ornate |
| 00154_sailboat |  | Boat     Sailboat     Ocean White     Wind |
| 00155_sandal |  | Footwear     Sandals     Leather Straps     Brown |

| Image Label | Test Image in ThingsEEG | Category-based label |
|---|---|---|
| 00156_sandpaper | | Tool   Sandpaper   Abrasive
Roll   Rough |
| 00157_sausage | | Food   Sausage   Sliced
Smoked   Meat |
| 00158_scallion | | Vegetable   Scallion   Green
Fresh   Bundle |
| 00159_scallop | | Seafood   Scallops   Seared
Plate   Garnish |
| 00160_scooter | | Vehicle   Scooter   Electric
Green   Urban |
| 00161_seagull | | Bird   Seagull   Beach
White   Walking |
| 00162_seaweed | | Marine   Seaweed   Underwater
Aquatic   Sunlight |

| Image Label | Test Image in ThingsEEG | Category-based label | | |
|---|---|---|---|---|
| 00163_seed |  | Food Brown | Seeds Spoon | Flax |
| 00164_skateboard |  | Sport Outdoor | Skateboard Deck | Wheels |
| 00165_sled |  | Winter Snow | Sled Sleigh | Wooden |
| 00166_sleeping_bag |  | Camping Outdoor | Sleeping Frost | Bag |
| 00167_slide |  | Playground Ladder | Slide Slide | Blue Outdoor |
| 00168_slingshot |  | Tool Rubber | Slingshot Y-shaped | Wooden |
| 00169_snowshoe |  | Footwear Running | Snowshoes Winter | Yellow |

*Continued on next page*

| Image Label | Test Image in ThingsEEG | Category-based label |
|---|---|---|
| 00170_spatula |  | Utensil   Spatula   Metal
Slotted   Handle |
| 00171_spoon |  | Utensil   Spoon   Metal
Reflection   Curved |
| 00172_station_wagon |  | Vehicle   Station   Wagon
Red   Classic |
| 00173_stethoscope |  | Medical   Stethoscope   Instrument
Black   Diagnosis |
| 00174_strawberry |  | Fruit   Strawberry   Red
Ripe   Plant |
| 00175_submarine |  | Vessel   Submarine   Navy
Water   Stealth |
| 00176_suit |  | Clothing   Suit   Formal
Business   Tailored |

| Image Label | Test Image in ThingsEEG | Category-based label |
|---|---|---|
| 00177_t-shirt |  | Clothing   T-shirt   White
Event   Hanger |
| 00178_table |  | Furniture   Table   Wooden
Square   Drawer |
| 00179_taillight |  | Vehicle   Taillight   Pink
Classic   Chrome |
| 00180_tape_recorder |  | Device   Recorder   Cassette
Vintage   Audio |
| 00181_television |  | Electronics   Television   CRT
Screen   Retro |
| 00182_tiara |  | Crown   Tiara   Gold
Jewels   Red |
| 00183_tick |  | Insect   Tick   Parasite
Skin   Tiny |
| | | *Continued on next page* |

| Image Label | Test Image in ThingsEEG | Category-based label |
|---|---|---|
| 00184_tomato_sauce | | Food   Sauce   Tomato
Pot   Red |
| 00185_tongs | | Utensil   Tongs   Metal
Grip   Kitchen |
| 00186_tool | | Tools   Hammer   Pliers
Screwdriver   Utility |
| 00187_top_hat | | Accessory   Top-hat   Cane
Gloves   Velvet |
| 00188_treadmill | | Exercise   Treadmill   Machine
Indoor   Fitness |
| 00189_tube_top | | Clothing   Top   Striped
Yellow   Knitted |
| 00190_turkey | | Bird   Turkey   Feathers
Fanned   Brown |

*Continued on next page*

| Image Label | Test Image in ThingsEEG | Category-based label |
|---|---|---|
| 00191_unicycle |  | Vehicle   Unicycle   Wheel
Tire     Seat |
| 00192_vise |  | Tool    Vise    Metal
Clamp   Adjustable |
| 00193_volleyball |  | Sport   Volleyball   Beach
Ball    Sand |
| 00194_wallpaper |  | Interior   Wallpaper   Pattern
Vintage   Wood |
| 00195_walnut |  | Food   Walnut   Nut
Shell   Brown |
| 00196_wheat |  | Crop   Wheat   Grain
Field   Stalk |
| 00197_wheelchair |  | Mobility   Wheelchair   Manual
Wheels   Seat |

| Image Label | Test Image in ThingsEEG | Category-based label |
|---|---|---|
| 00198_windshield | | Vehicle    Windshield    Glass
Car          Street |
| 00199_wine | | Beverage    Wine    Glass
Grapes       Red |
| 00200_wok | | Cookware    Wok    Pan
Handles    Black |

## B   THE IMAGE GENERATION RESULTS OF NECOMIMI

In this section, we will present all the images generated by various EEG encoders within the NECOMIMI framework using a fixed random seed. These images are generated using the testing set of the ThingsEEG dataset in a zero-shot setting, meaning that the model has not seen these categories during the EEG-Image contrastive learning training process. All the images illustrate the progression of visual representations generated using different embedding techniques in a diffusion model: (a) Top row: The original images shown to subjects (ground truth). (b) Second row: Images generated by the CLIP-ViT embeddings of the original images. It is only related to the seed and has nothing to do with the subject and EEG encoder. (c) Third row: Images generated by one-stage method using pure EEG embeddings with the EEG encoder. (d) Fourth row: Images generated by two-stage NECOMIMI method using pure EEG embeddings with EEG encoder.

### B.1   USING NICE AS THE EEG ENCODER

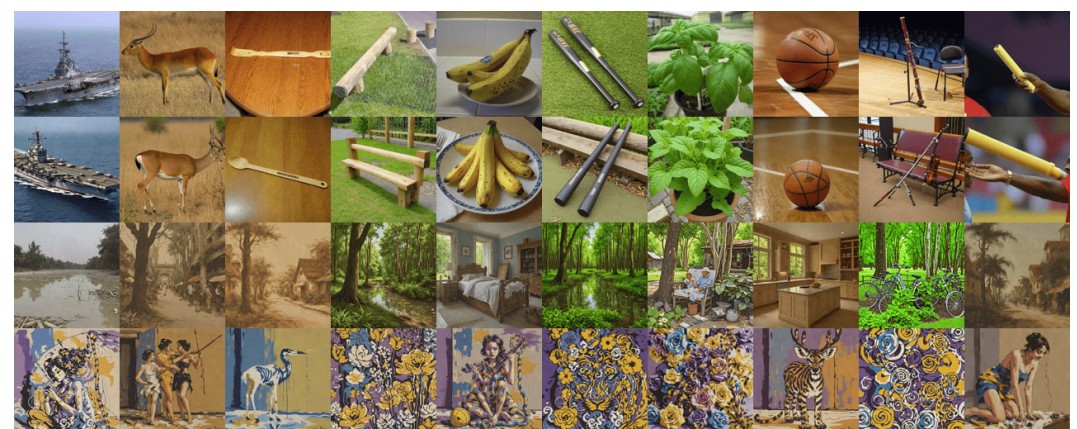

Figure 5: Random selected generated images in Subject 6 with NICE EEG encoder.

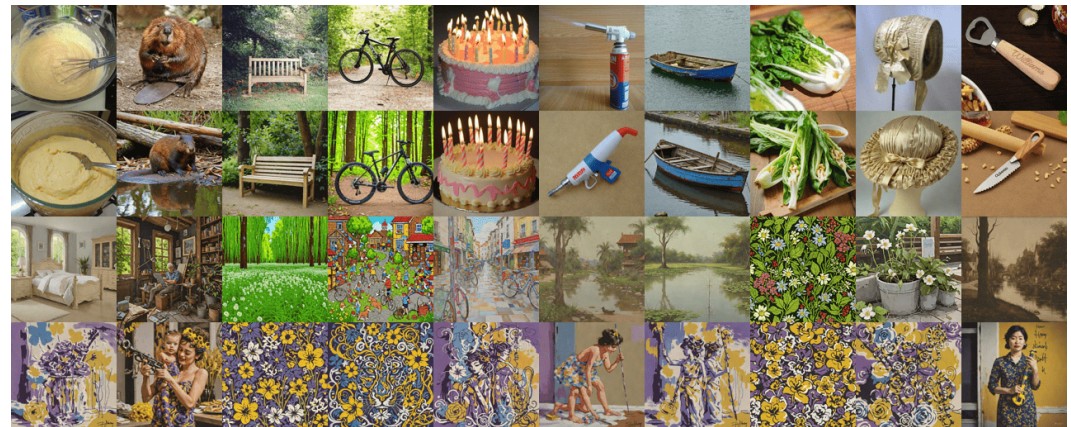

Figure 6: Random selected generated images in Subject 6 with NICE EEG encoder.

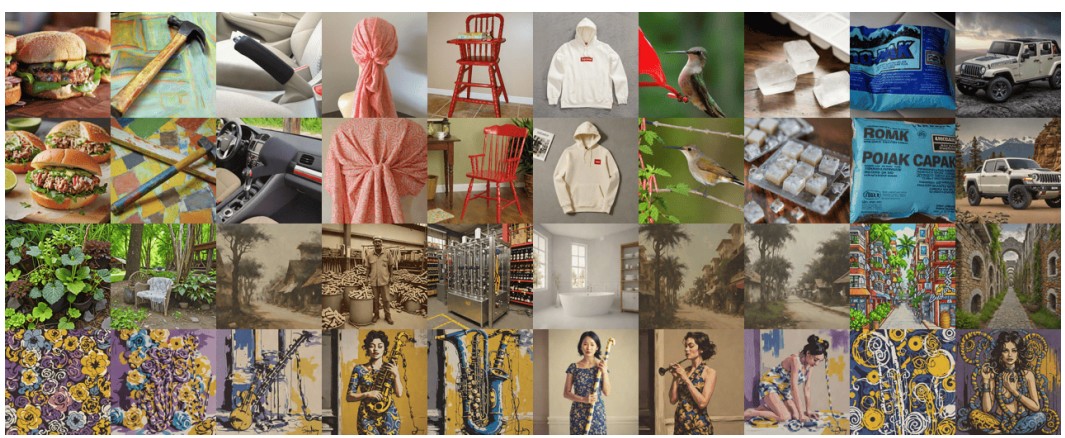

Figure 7: Random selected generated images in Subject 6 with NICE EEG encoder.

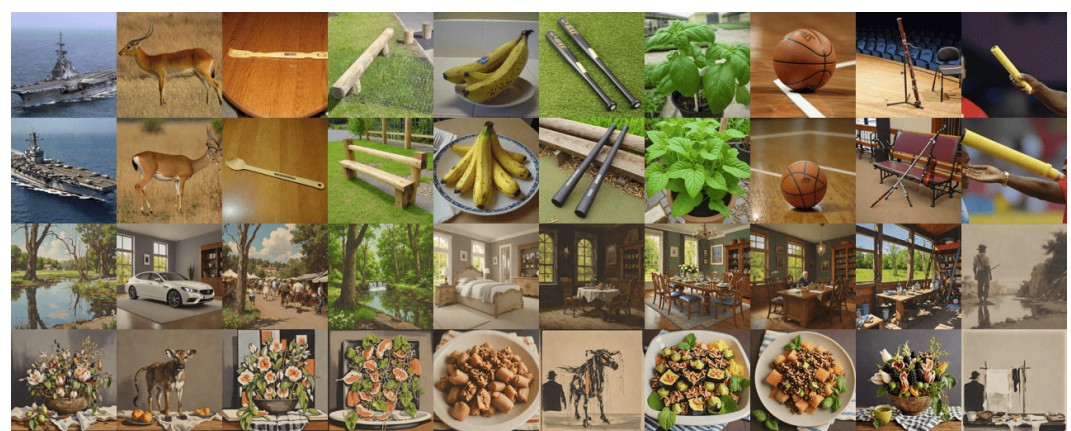

Figure 8: Random selected generated images in Subject 7 with NICE EEG encoder.

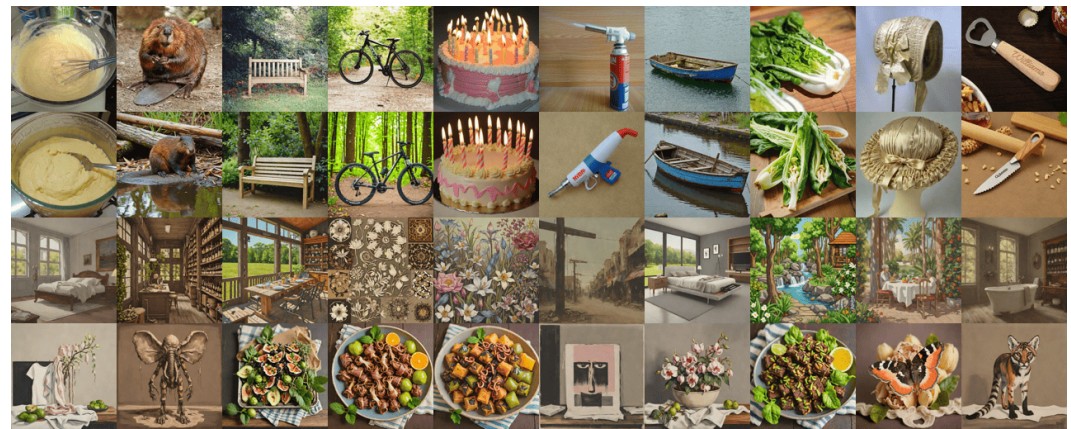

Figure 9: Random selected generated images in Subject 7 with NICE EEG encoder.

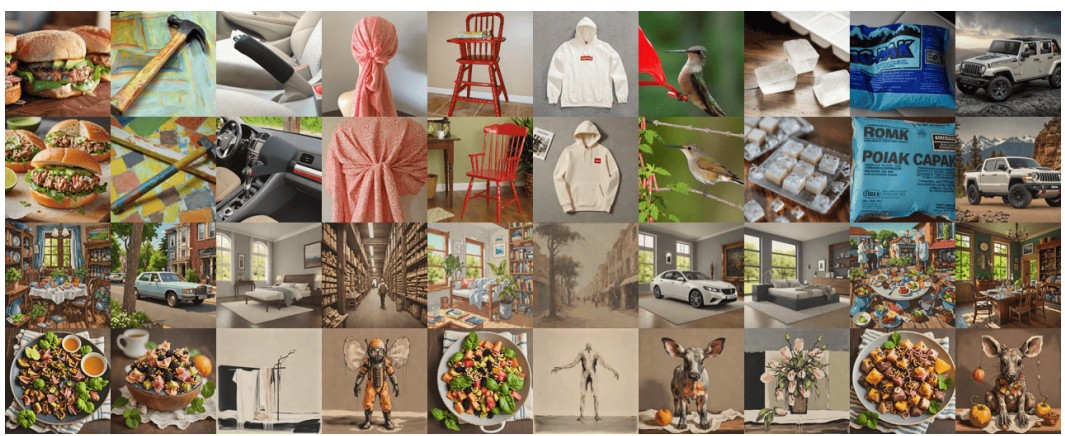

Figure 10: Random selected generated images in Subject 7 with NICE EEG encoder.

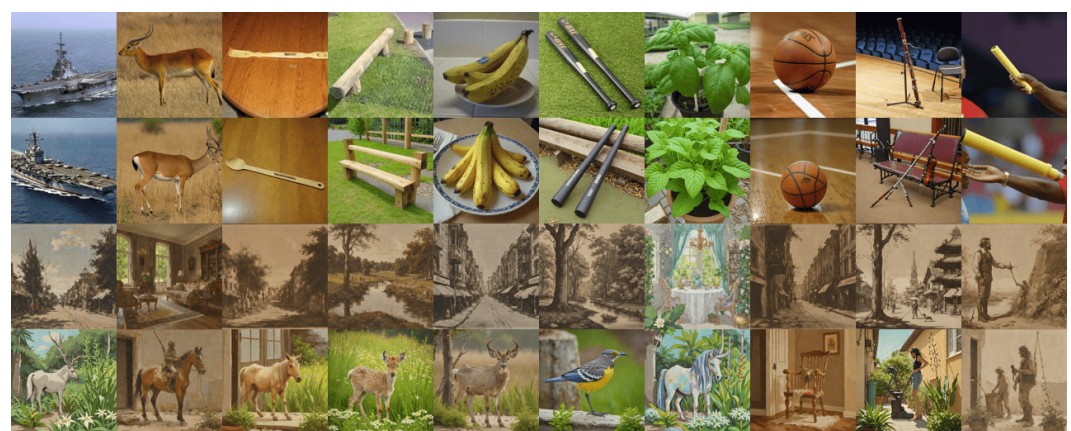

Figure 11: Random selected generated images in Subject 8 with NICE EEG encoder.

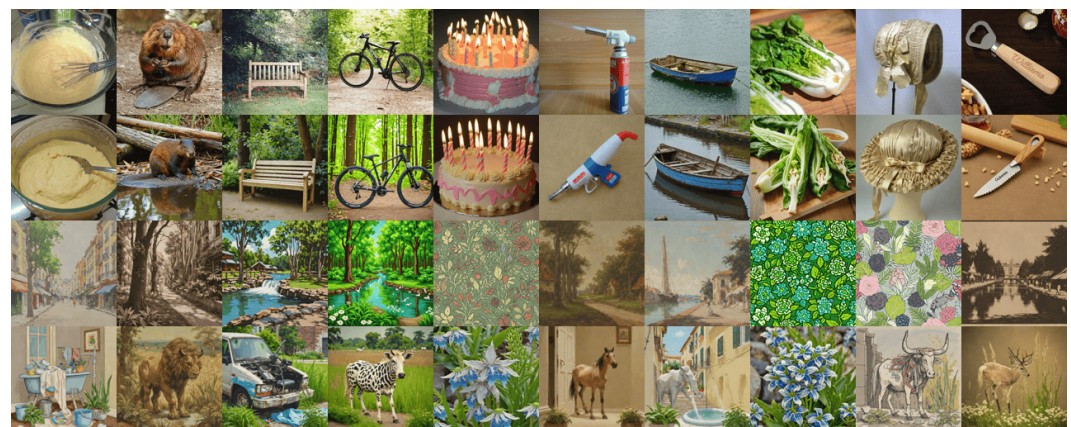

Figure 12: Random selected generated images in Subject 8 with NICE EEG encoder.

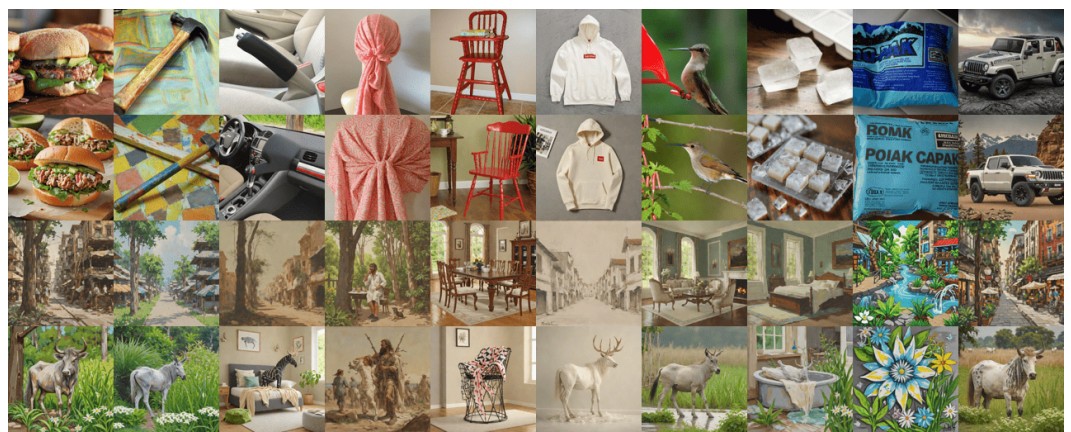

Figure 13: Random selected generated images in Subject 8 with NICE EEG encoder.

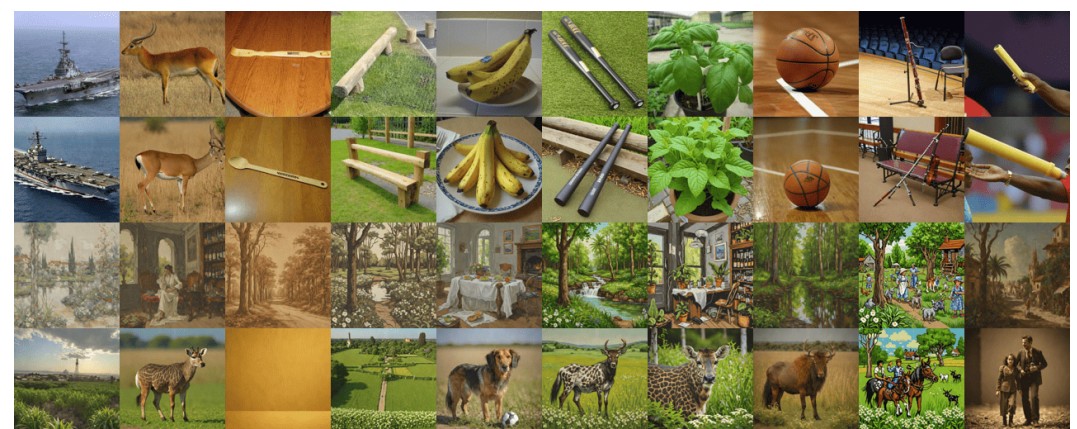

Figure 14: Random selected generated images in Subject 6 with Nervformer EEG encoder.

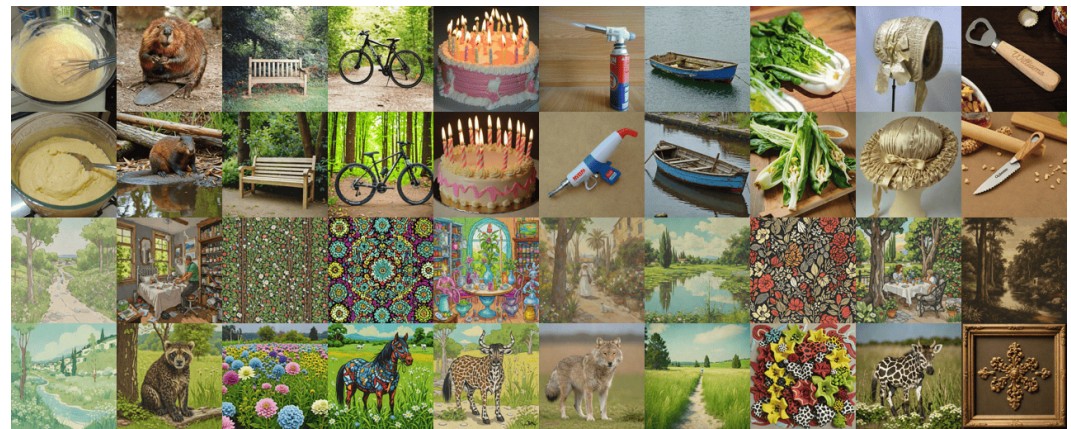

Figure 15: Random selected generated images in Subject 6 with Nervformer EEG encoder.

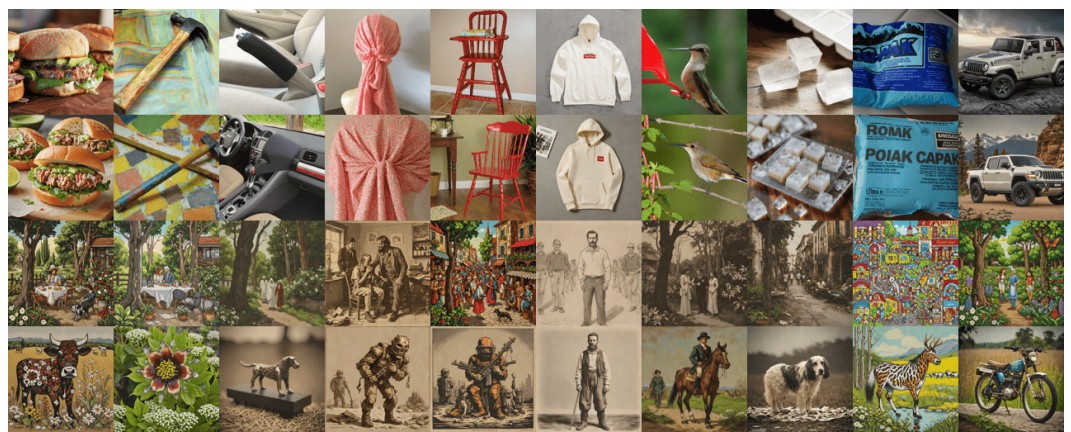

Figure 16: Random selected generated images in Subject 6 with Nervformer EEG encoder.

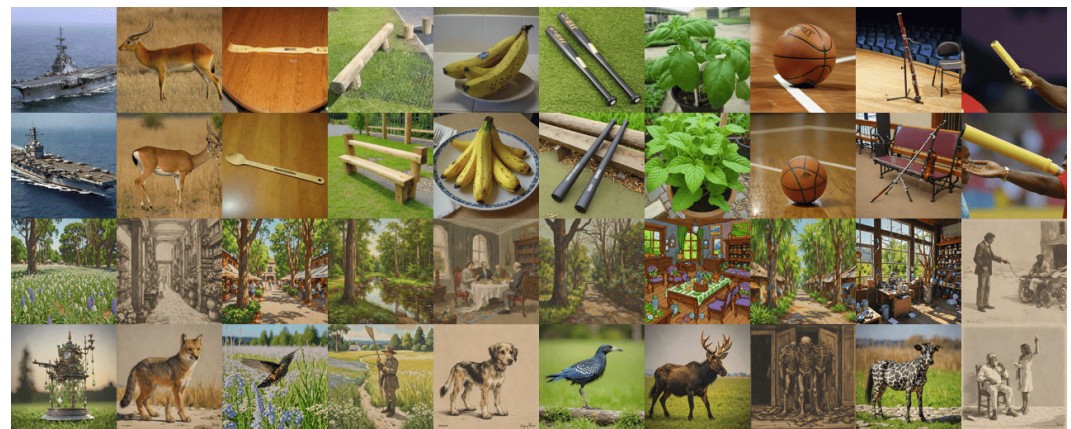

Figure 17: Random selected generated images in Subject 7 with Nervformer EEG encoder.

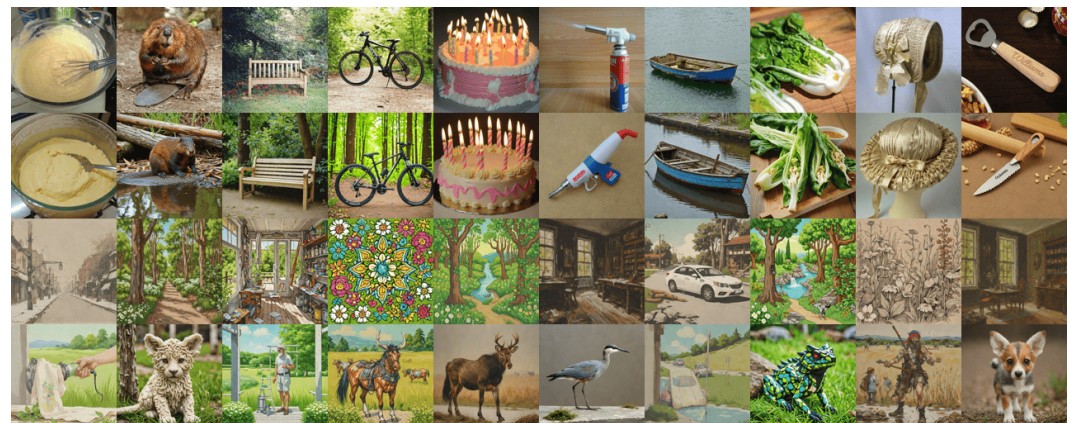

Figure 18: Random selected generated images in Subject 7 with Nervformer EEG encoder.

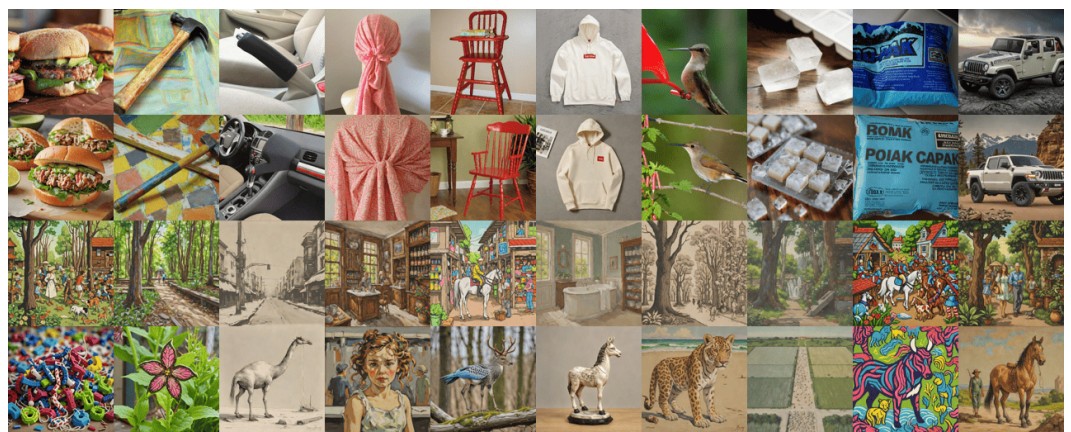

Figure 19: Random selected generated images in Subject 7 with Nervformer EEG encoder.

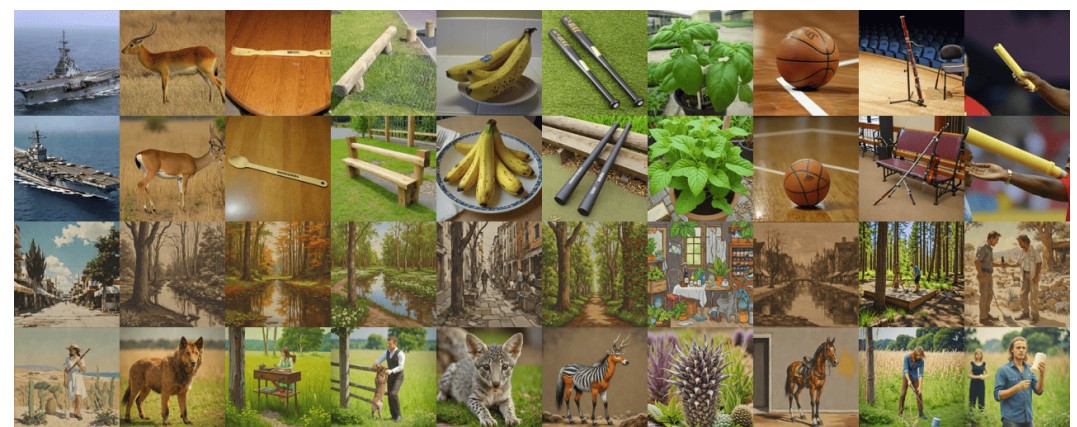

Figure 20: Random selected generated images in Subject 8 with Nervformer EEG encoder.

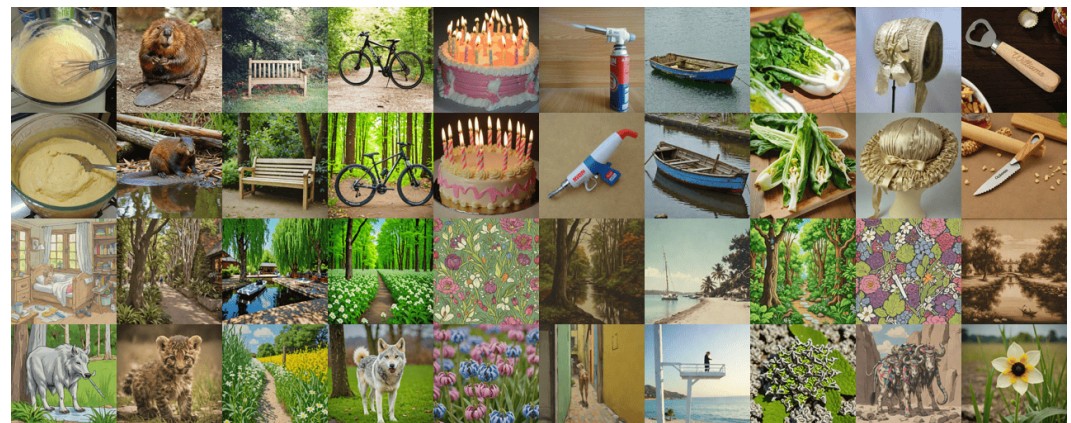

Figure 21: Random selected generated images in Subject 8 with Nervformer EEG encoder.

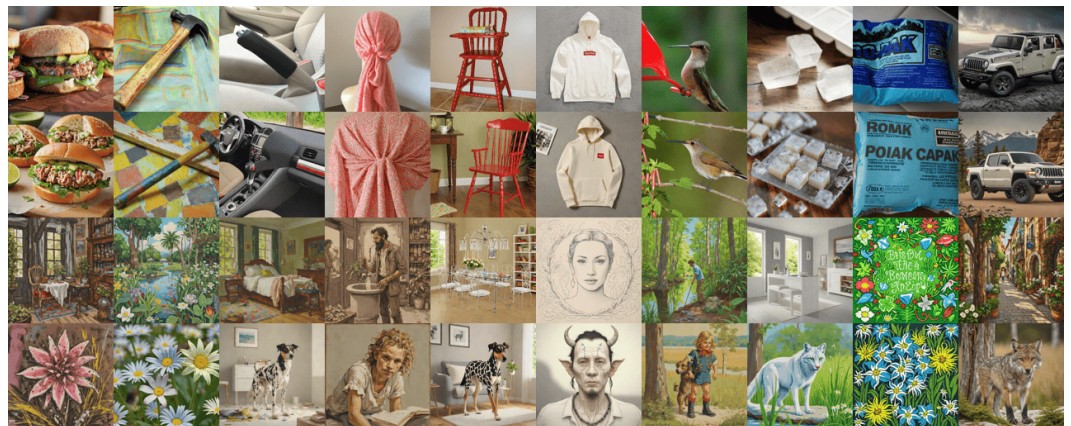

Figure 22: Random selected generated images in Subject 8 with Nervformer EEG encoder.

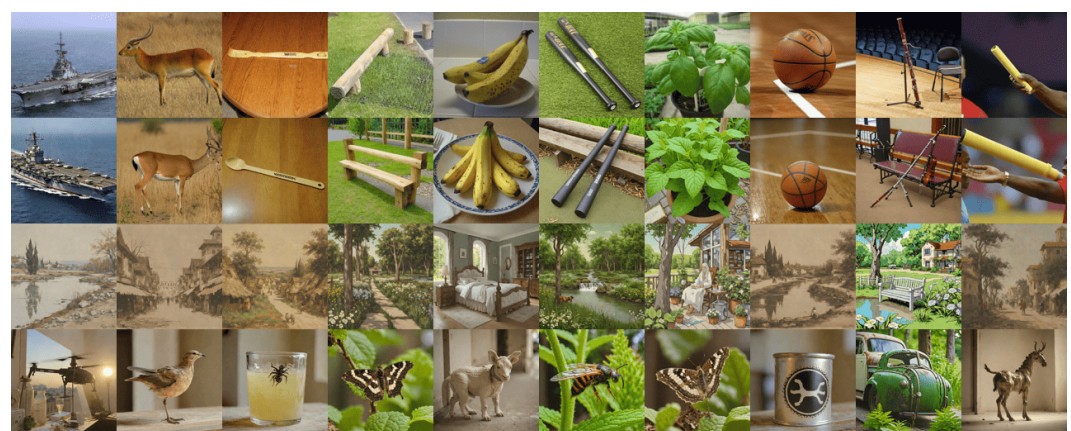

Figure 23: Random selected generated images in Subject 6 with MUSE EEG encoder.

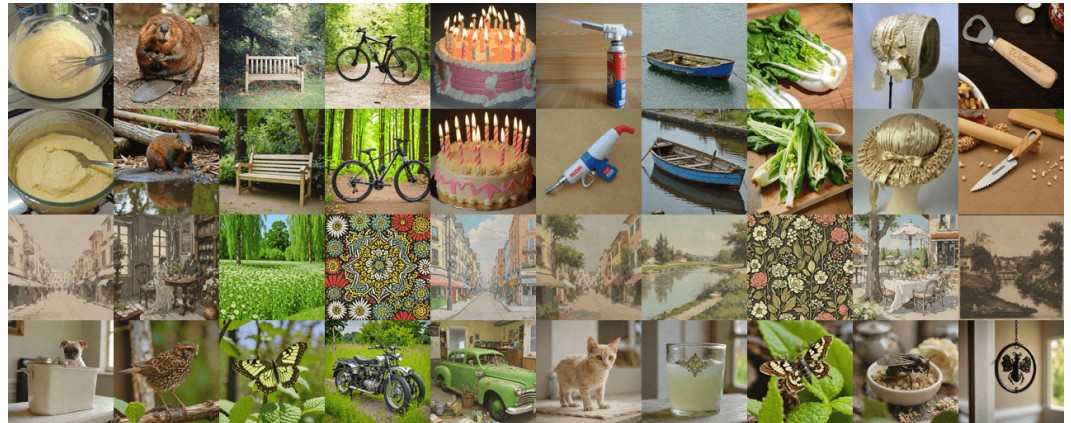

Figure 24: Random selected generated images in Subject 6 with MUSE EEG encoder.

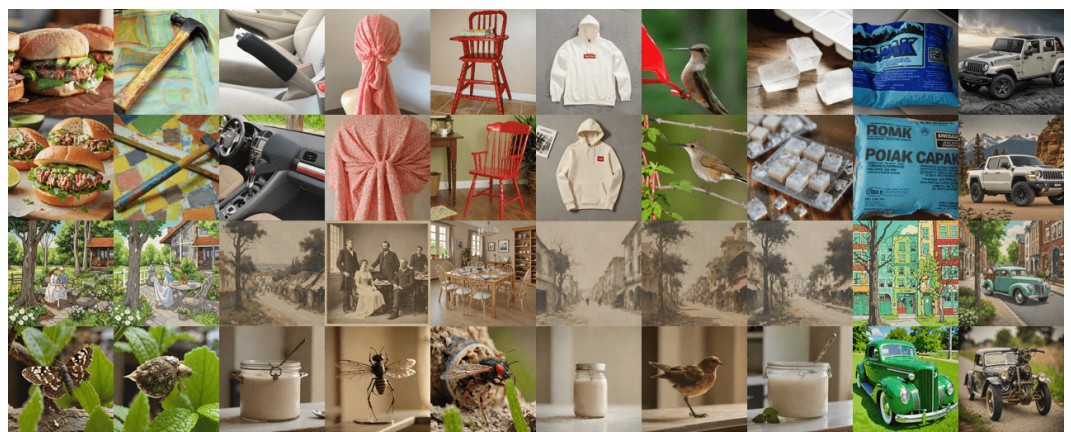

Figure 25: Random selected generated images in Subject 6 with MUSE EEG encoder.

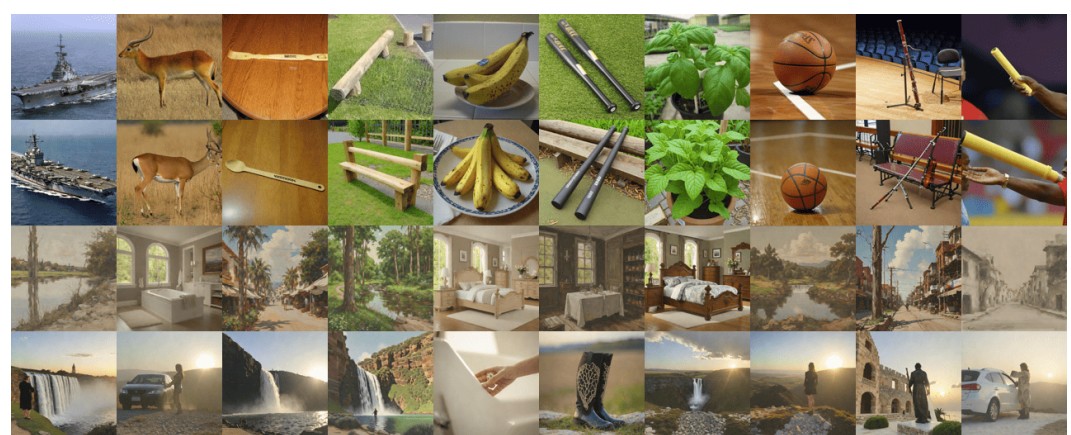

Figure 26: Random selected generated images in Subject 7 with MUSE EEG encoder.

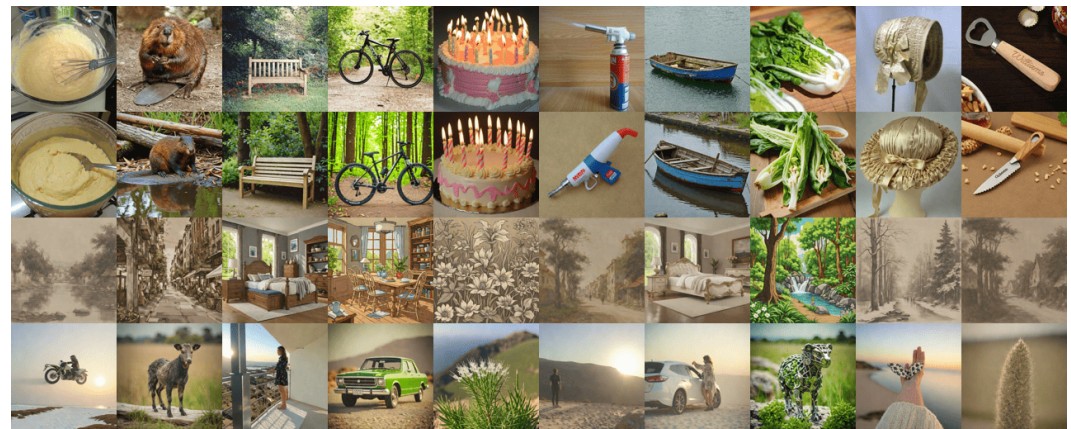

Figure 27: Random selected generated images in Subject 7 with MUSE EEG encoder.

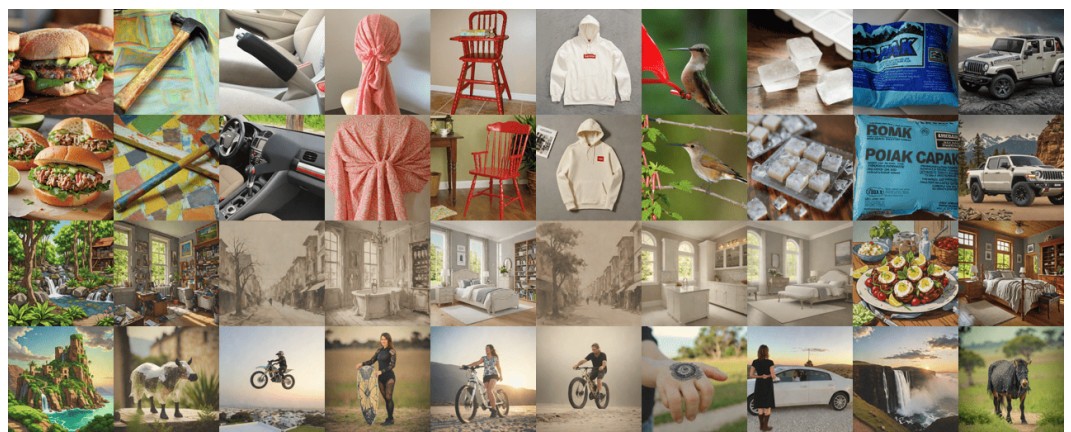

Figure 28: Random selected generated images in Subject 7 with MUSE EEG encoder.

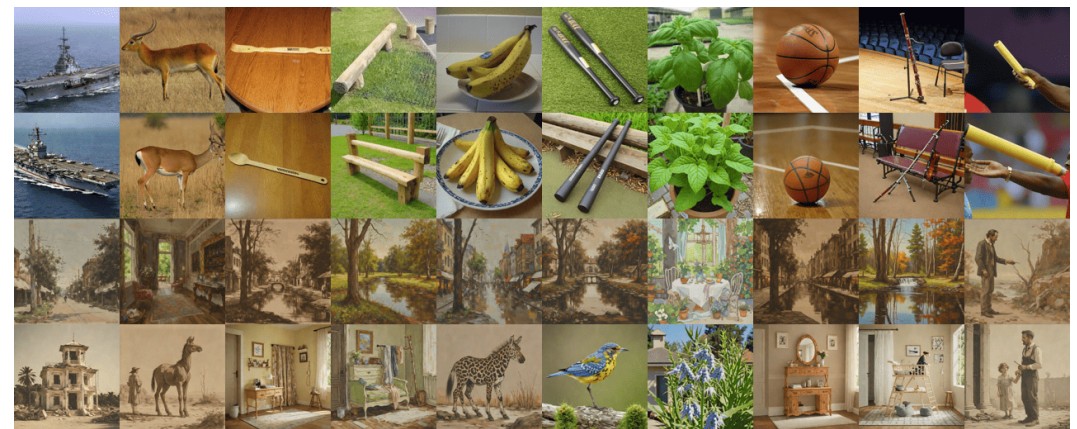

Figure 29: Random selected generated images in Subject 8 with MUSE EEG encoder.

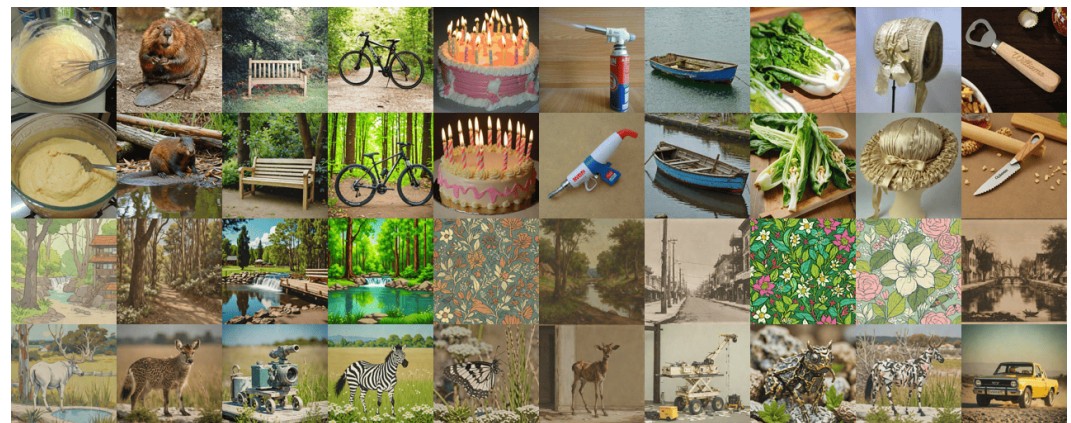

Figure 30: Random selected generated images in Subject 8 with MUSE EEG encoder.

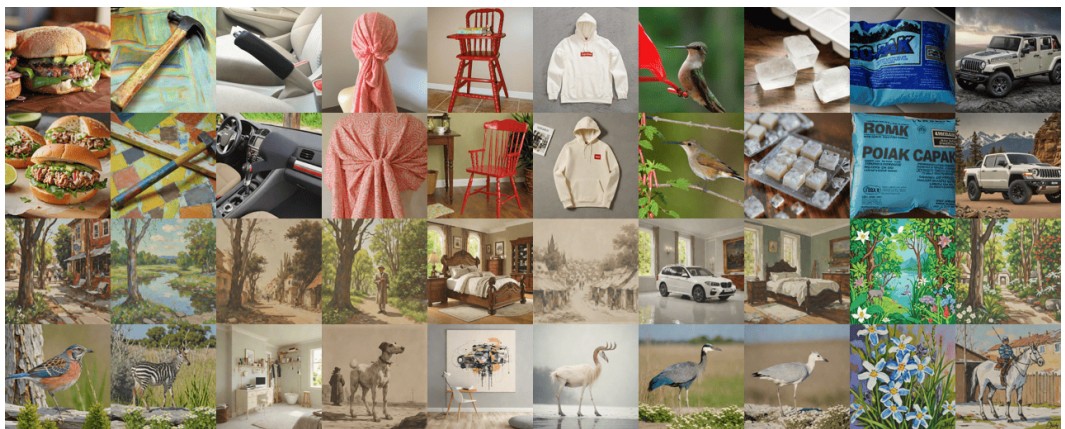

Figure 31: Random selected generated images in Subject 8 with MUSE EEG encoder.

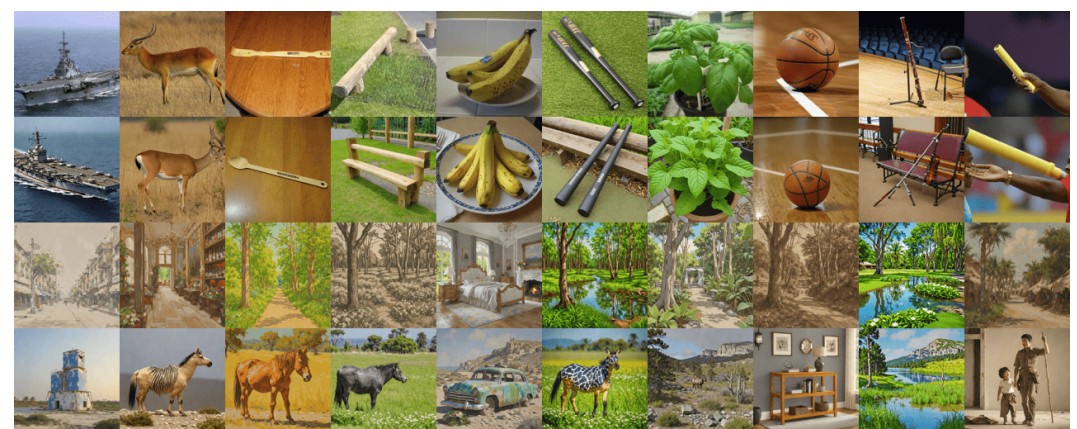

Figure 32: Random selected generated images in Subject 6 with ATM-S EEG encoder.

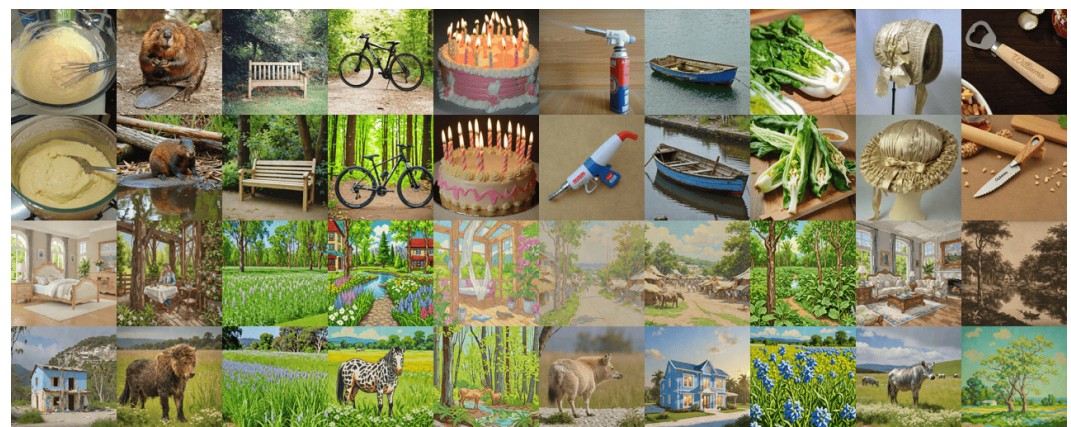

Figure 33: Random selected generated images in Subject 6 with ATM-S EEG encoder.

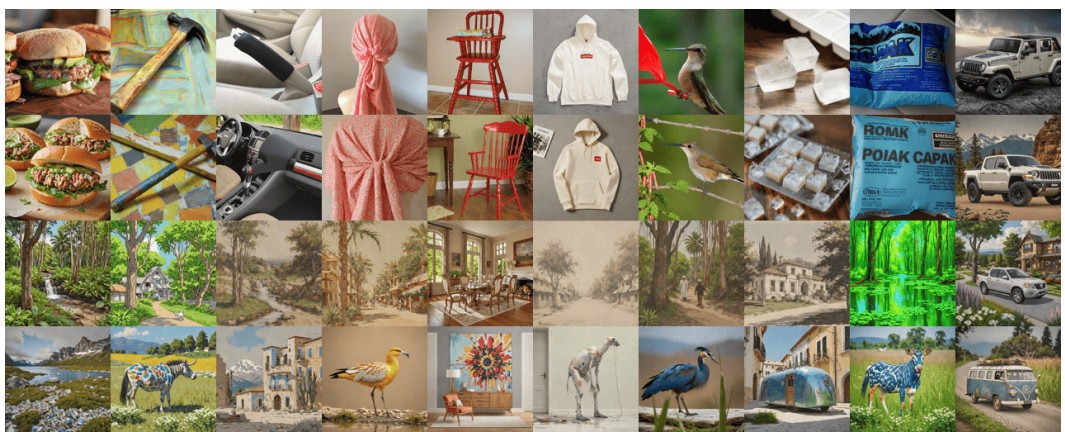

Figure 34: Random selected generated images in Subject 6 with ATM-S EEG encoder.

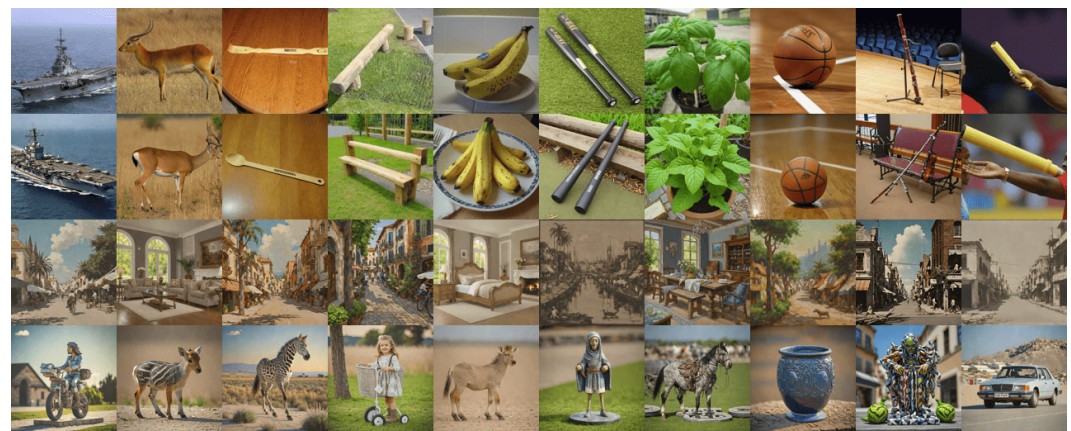

Figure 35: Random selected generated images in Subject 7 with ATM-S EEG encoder.

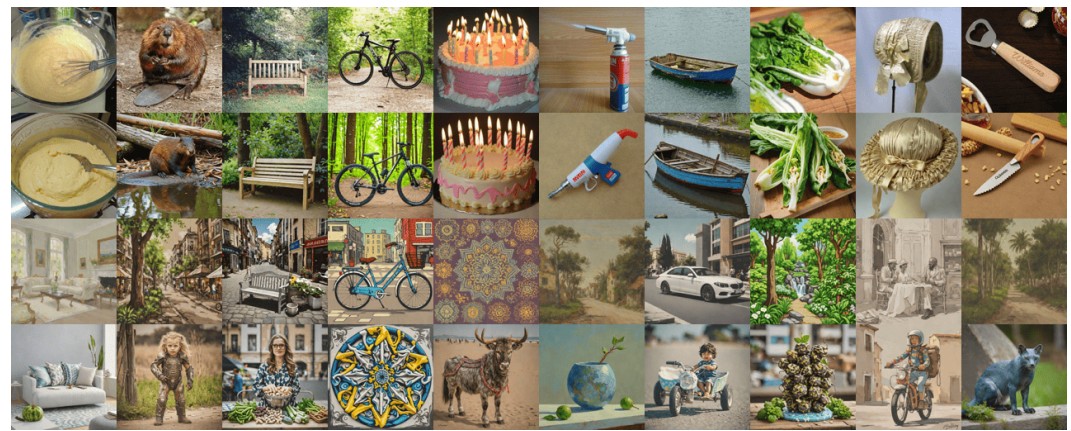

Figure 36: Random selected generated images in Subject 7 with ATM-S EEG encoder.

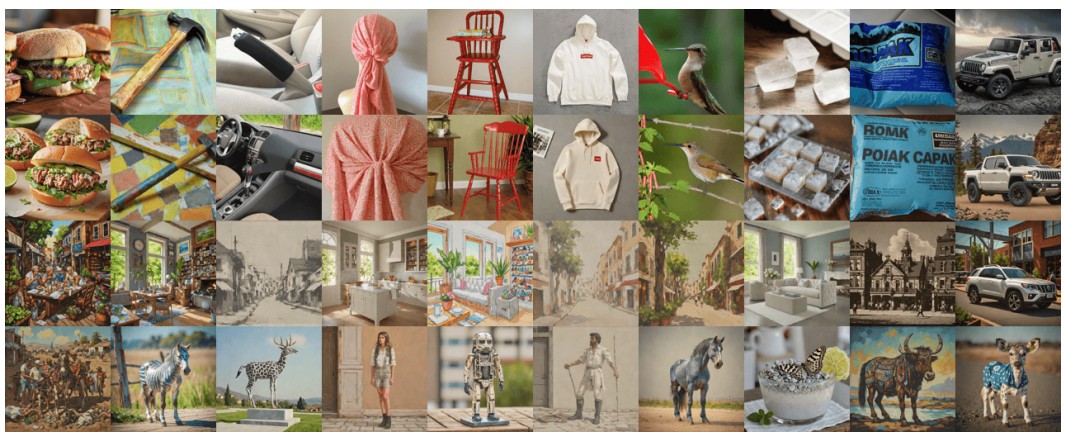

Figure 37: Random selected generated images in Subject 7 with ATM-S EEG encoder.

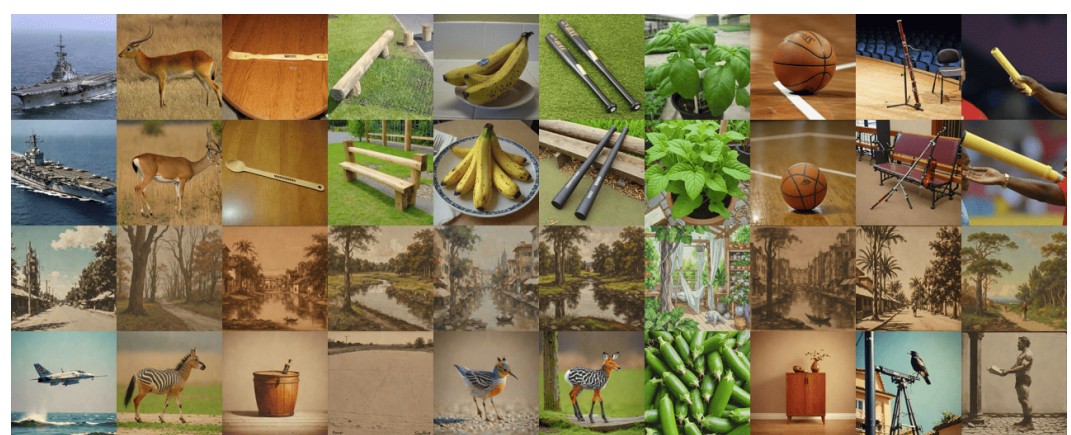

Figure 38: Random selected generated images in Subject 8 with ATM-S EEG encoder.

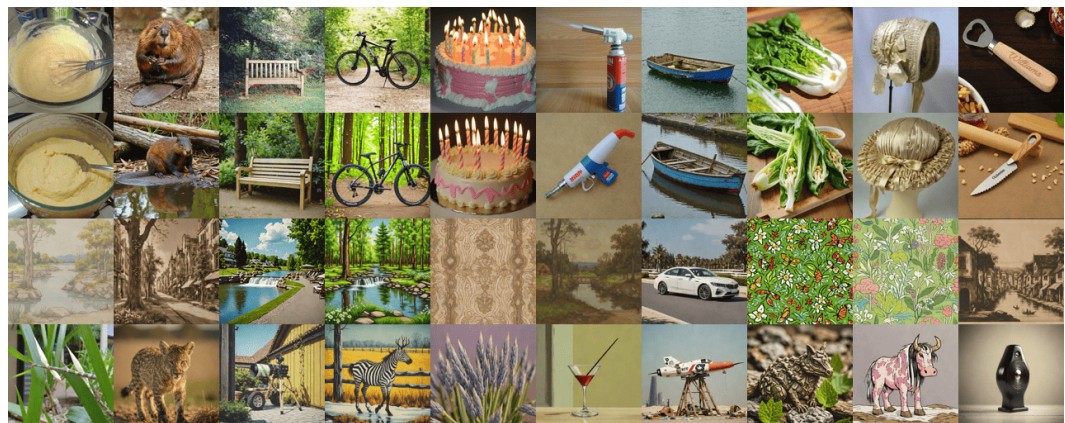

Figure 39: Random selected generated images in Subject 8 with ATM-S EEG encoder.

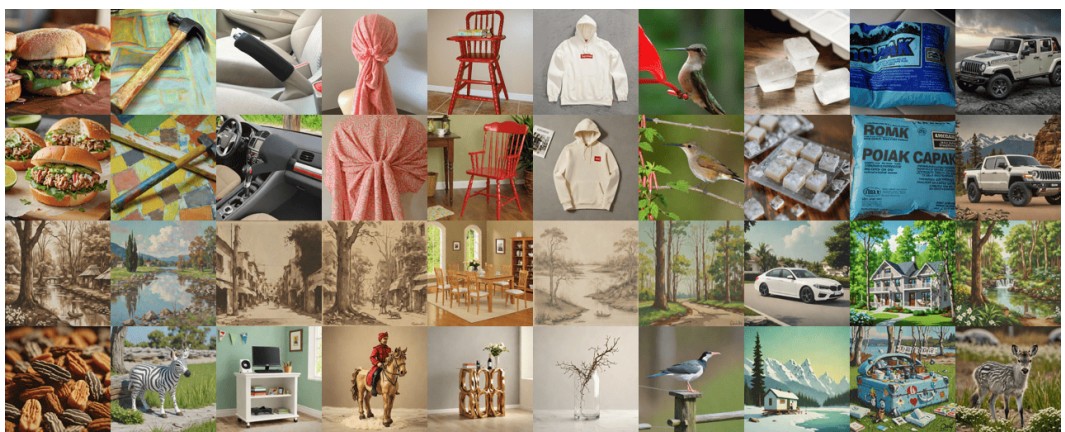

Figure 40: Random selected generated images in Subject 8 with ATM-S EEG encoder.

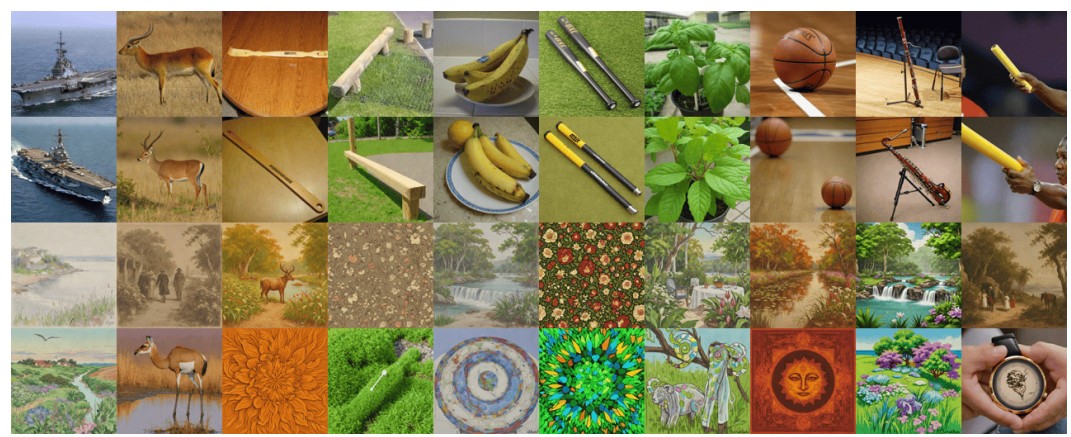

Figure 41: Random selected generated images in Subject 6 with NERV EEG encoder.

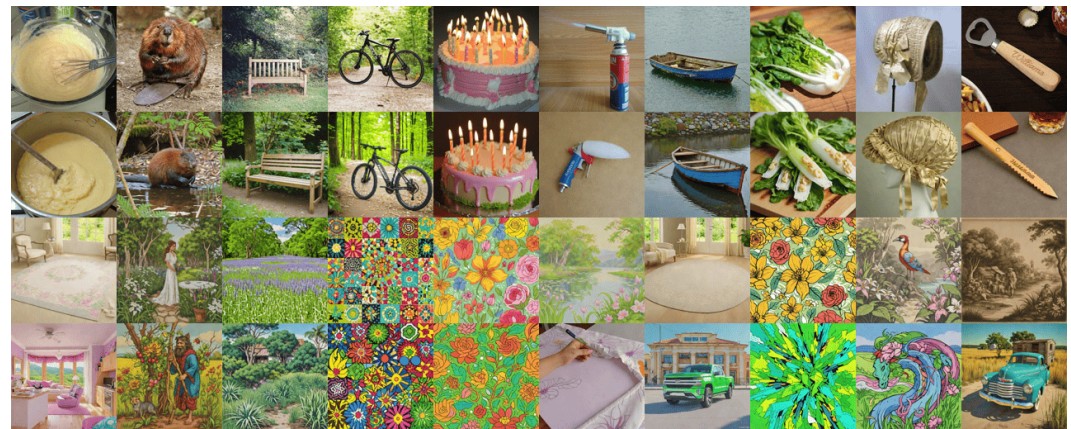

Figure 42: Random selected generated images in Subject 6 with NERV EEG encoder.

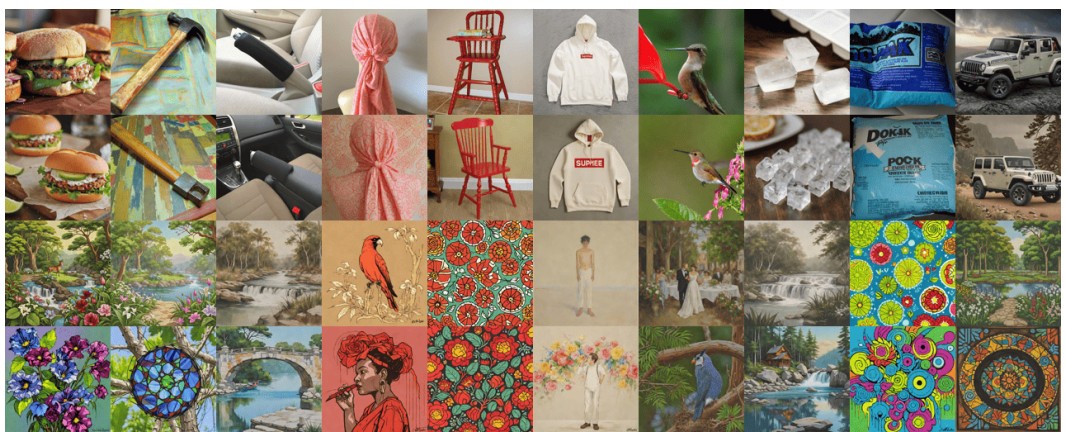

Figure 43: Random selected generated images in Subject 6 with NERV EEG encoder.

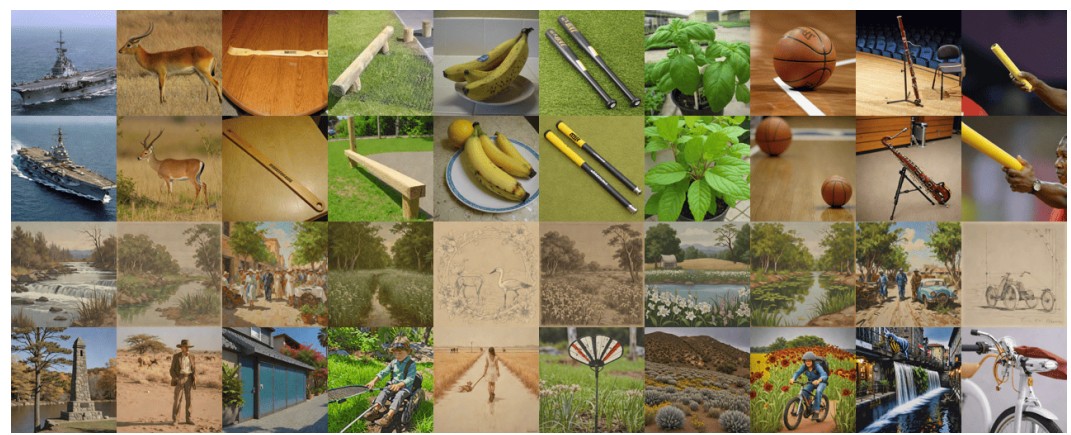

Figure 44: Random selected generated images in Subject 7 with NERV EEG encoder.

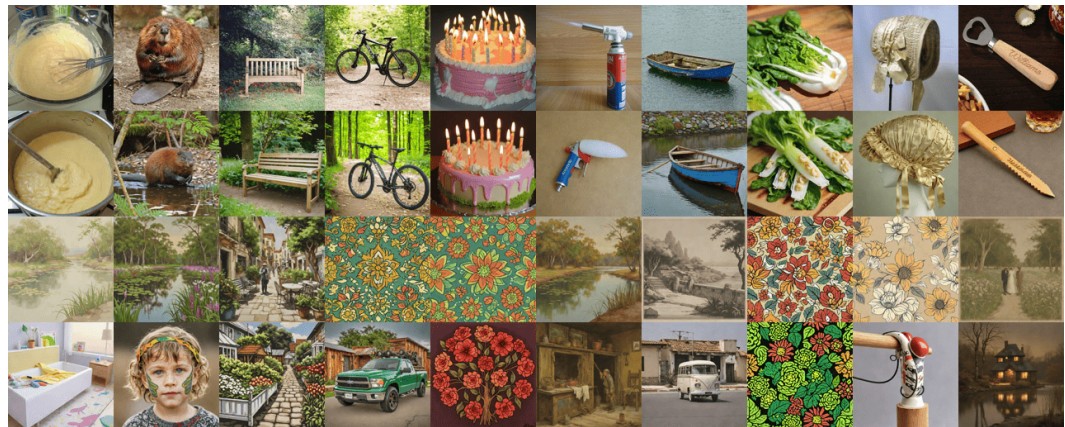

Figure 45: Random selected generated images in Subject 7 with NERV EEG encoder.

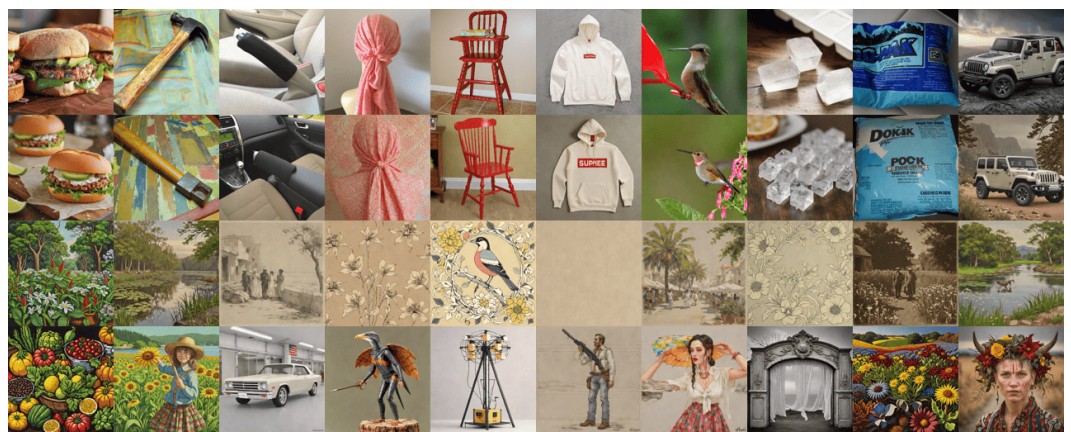

Figure 46: Random selected generated images in Subject 7 with NERV EEG encoder.

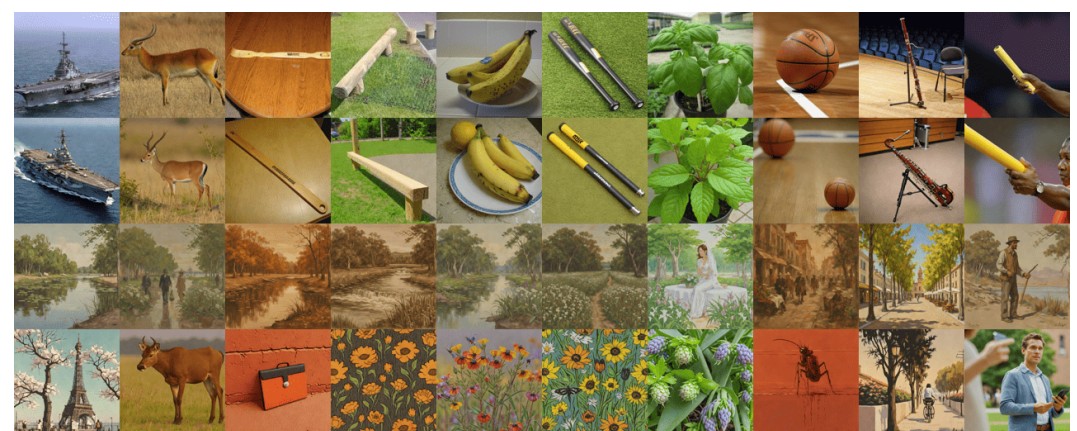

Figure 47: Random selected generated images in Subject 8 with NERV EEG encoder.

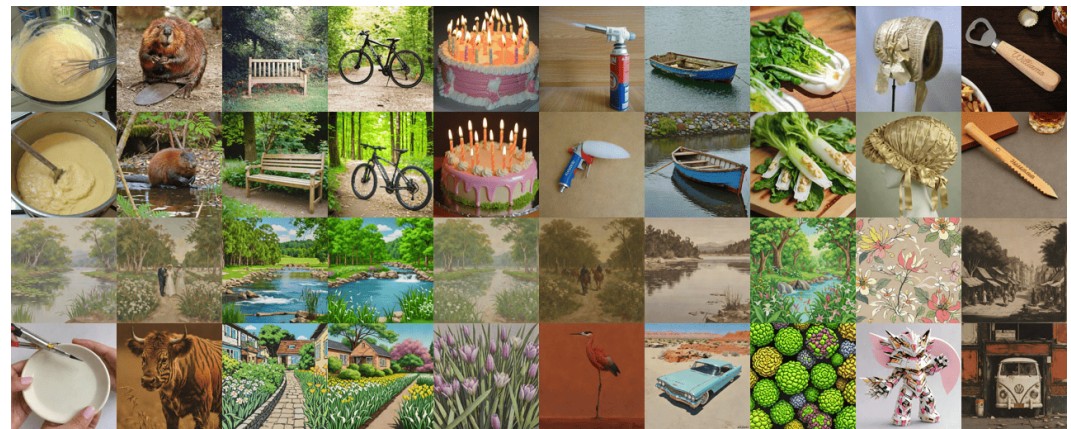

Figure 48: Random selected generated images in Subject 8 with NERV EEG encoder.

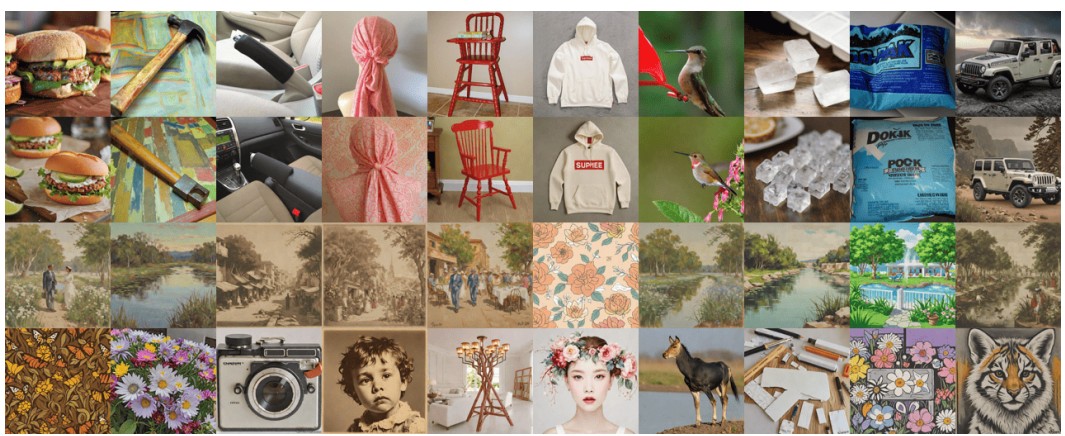

Figure 49: Random selected generated images in Subject 8 with NERV EEG encoder.

