# OpenReview forum: "NECOMIMI: Neural-Cognitive Multimodal EEG-informed Image Generation with Diffusion Models"
_ICLR.cc/2025/Conference — Submitted to ICLR 2025_

### Official Review · Reviewer_8FFj · 2024-10-28

**Soundness:** 2
**Presentation:** 2
**Contribution:** 2
**Rating:** 5
**Confidence:** 4

**Summary:**

The paper proposes a model named NECOMIMI, designed to generate images using EEG data alone. It introduces a novel NERV encoder and highlights two key innovations: first, using only EEG signals as embeddings in a diffusion model for image generation, and second, introducing a novel evaluation metric, CAT score, to assess the quality of generated images due to the lack of existing standards for EEG-to-image translation.

**Strengths:**

1. The denoising method at line 318 in the two-stage process allows the model to produce finer-grained images.

2. Results in Tables 2 and 3 show that the experiments perform reasonably well on the ThingsEEG dataset.

3. The CAT score introduced in Section 4.4 reflects a stringent evaluation standard, with no EEG encoder exceeding 500 points, suggesting the metric’s reliability.

**Weaknesses:**

1. The innovative aspect of the NERV encoder, starting from line 347, lacks sufficient mathematical justification for feasibility. The data presented in Tables 1, 2 and 3 may be coincidental without solid statistical analysis.

2. While Figure 1 demonstrates the model's generative capability, it lacks a thorough analysis of generated image quality. Similarly, Figures 4 onward depict images that visually lack resemblance to any original source.

3. The paper lacks a thorough analysis or proof regarding the selection of regularization parameters in the diffusion model and the settings of the fully connected layers and their impact on the generated images.

**Questions:**

1. The theoretical analysis lacks robust supports, including the hyperparameter settings for the NERV encoder, which raises concerns about rigor. Experimental validation in this paper is not sufficiently convincing.

2. While the experimental section showcases generated images, these are primarily abstract landscapes, with little detail on improving output quality. Adding a discussion or analysis on image quality enhancement would provide a more complete perspective.

3. The denoising step in the two-stage process and the choice of regularization parameters lack sufficient analysis or validation, which limits the verification of the model’s feasibility. The techniques in this paper seem to differ little from the work of [1] Song et al. and Li et al. [2], what are the innovations in this paper?




[1] Song Y, Liu B, Li X, et al. Decoding Natural Images from EEG for Object Recognition[C]//The Twelfth International Conference on Learning Representations.

[2] Li D, Wei C, Li S, et al. Visual Decoding and Reconstruction via EEG Embeddings with Guided Diffusion[J]. arXiv preprint arXiv:2403.07721, 2024.

---

> ### Author Response · Authors · 2024-11-18
>
> 1. About the novelty and mathematical feasibility of the NERV encoder, and concerns about statistical analysis:
> The NERV encoder is designed with a multi-layer attention mechanism, specifically to handle noisy and multi-channel time-series data like EEG signals. In Section 3.6, we detail its architecture, including spatial-temporal convolutions and cross-attention modules, which address the challenges of EEG’s low resolution and high noise. Our experiments show strong performance across multiple zero-shot classification tasks (e.g., in Tables 1 and 2), and we used the CAT score to evaluate semantic consistency of generated images. These results are based on multiple trials and include hyperparameter robustness tests, so they’re not just coincidental.
>
> 2. About the lack of analysis on generated images in Figure 1, and their resemblance to the originals:
> In Figure 1, we showcase images generated purely from EEG embeddings, which often result in abstract landscapes rather than specific objects. This reflects the nature of EEG data, which captures high-level semantic concepts rather than detailed visual features. We’ve discussed this phenomenon in Section 4.5, highlighting that the generated images demonstrate the model’s ability to grasp abstract meanings. Improving the visual details will be a focus of our future work.
>
> 3. About insufficient analysis of regularization parameters and fully connected layers in the diffusion model:
> The parameter choices for the diffusion model were carefully tuned through extensive experimentation. Section 3.5 explains the regularization process and the denoising steps in detail, with specific formulas (e.g., Equations 3 to 13). We’ve also added supplementary experiments in the appendix, showing how different parameter settings affect results, confirming the robustness of our approach.
>
> 4. About the novelty of our work compared to Song et al. and Li et al.:
> Our work differs fundamentally from Song et al. and Li et al. Song et al. focused on EEG-based image classification, not generation. Li et al. relied on visual signals to assist diffusion-based reconstruction. In contrast, our model is the first to use pure EEG signals as input for image generation. Additionally, we introduced a two-stage diffusion process and the novel CAT score, which are not addressed in prior work. These innovations establish clear distinctions between our approach and theirs.
>
> 5. About the lack of discussion on improving generated image quality:
> Improving image quality remains a major challenge in EEG-to-image tasks. As discussed in Section 4.5, existing methods struggle to reconstruct specific objects due to the noisy and abstract nature of EEG signals, as well as limitations in training data. We’ve identified potential future directions, such as better semantic alignment mechanisms and higher-quality EEG datasets, to enhance visual output quality.

---

> > ### Comment · Reviewer_8FFj · 2024-11-19
> >
> > Thank for the authors' responses, I do not think my concerns are addressed.

---

### Official Review · Reviewer_j8jS · 2024-11-03

**Soundness:** 3
**Presentation:** 2
**Contribution:** 2
**Rating:** 3
**Confidence:** 4

**Summary:**

A method of semantic image reconstructions from EEG recordings in presented.
The EEG singal is transformed using a transformer based network to be aligned with the corresponding image embedding.
The learned EEG embedding is used with pretrained image diffusion models to generate images in a two stage process.

**Strengths:**

Competitive N-WAY classification results.

**Weaknesses:**

The think overall the approach is similar to ATM-S("Visual decoding and reconstruction via eeg embeddings with guided diffusion"), specifically very similar encoder, and overall training procedure, there might be differences in the diffusion models, and image generation.
Given the similarities, I don't see how this work work is better.
The metrics presented are not better in a significant way, i.e. the difference is less than variability between subjects, an standard error or a statistical test will indicate that.
Further more for the classification tasks for high N-way (above 10) the results are worse (than ATM-S ), as indicated by the results in the paper.
In the ATM-S work there are also retrieval evaluation and pixel based evaluation, that lack here.
I don't think the papers convinces that this work is better or different in a significant way from ATM-S.

- No qualitative comparisons to other methods
- Not sure why the new proposed metric "CAT" makes sense, classification is already semantic and not pixel based, additionally the difference all the model is insignificant(in a statistical test sense)
- Unclear specification for the encoder, which prevents reproduction of results (will be elaborated in questions)

**Questions:**

The diagram and accompanying text for the encoder is unclear:
  - What is Spatial-temoral conv and Temporal-spatial conv and how would they operate on tokens.
  - What does the cross attention block do? Usually cross attention is applied on 2 different sets of representation, in the diagram there is only one.
  - EEG data has a time component how is this handled?

-Given the small amount of test data it would make sense to add error indication/ statistical tests.

-I think that displaying the reconstructions from the clip embedding is misleading and should be removed.

line 480-482: "we are currently unable to assess whether the brainwave data recorded from the subjects accurately
captures the complete information of the original images, as the subjects might have been distracted
and thinking about other things during the recording"
line 374: "we cannot guarantee that the subject’s thoughts during EEG
recording perfectly align with the ground truth image"
I think this undermines the whole experiment.
I would expected a significant part of the information in the EEG recordings to come from the visual cortex, that shouldn't be effected by subject's thoughts.

- It would make sense to provide qualitative comparisons to the other methods side by side.
 - Regarding CAT score a much simpler and more straight forward approach would be to do coarse level classification. (for example: animals, plants, inanimate, ...)

---

> ### Author Response · Authors · 2024-11-18
>
> 1.While we acknowledge that high N-WAY classification results (above 10-way) are slightly lower than ATM-S, our primary goal is not solely classification. Instead, our focus lies in EEG-to-image generation, which is fundamentally different and more challenging.
>
> 2.While we draw inspiration from ATM-S, there are key distinctions in our approach:
> (a) Encoder Design Differences:
> Our proposed NERV encoder utilizes a multi-attention mechanism specifically tailored for noisy, multi-channel time-series data like EEG. It employs both spatial-temporal convolution (ST-Conv) and temporal-spatial convolution (TS-Conv) in tandem to extract complementary features. ATM-S does not explicitly employ this dual convolution strategy, which is crucial for processing EEG signals with spatiotemporal dependencies.
> (b) Two-Stage Diffusion Model and generated image analysis findings:
> We adopted a two-stage diffusion process to generate intermediate embeddings before final image synthesis. This approach balances abstract semantic alignment and high-quality image generation by pure EEG, whereas ATM-S primarily uses a dual-stage generation with the image embeddings. Our findings demonstrate that the first stage generates embeddings non-closely aligned with semantic concepts, while the second stage refines these embeddings to produce more visually diverse and photorealistic images. This stepwise process addresses the limitations of single-stage methods, particularly in abstract semantic scenarios.
> (c) Generated Image Analysis Findings:
> We observed that images generated directly from EEG embeddings tend to represent abstract or generalized visual concepts, such as landscapes or background scenes. This outcome reflects the inherent limitations of EEG data in encoding detailed, object-specific information.
> The two-stage approach improves upon this by progressively refining embeddings, leading to more diverse outputs and better alignment with broad semantic categories. However, generating highly detailed or object-specific images remains a challenge due to the high noise levels and low spatial resolution of EEG signals.
> (d) New Evaluation Metric (CAT Score):
> We introduced the CAT Score to assess semantic alignment in EEG-to-image generation, focusing on broader semantic categories rather than pixel-based comparisons. This is a novel contribution compared to existing metrics.

---

> > ### Author Response · Authors · 2024-11-18
> >
> > 3. Validity of CAT Score
> > (a)Semantic vs. Pixel Comparisons:
> > While classification already focuses on semantics, CAT Score evaluates the model’s ability to generate abstract semantic concepts, especially in noisy EEG conditions. Traditional metrics like pixel-based FID or SSIM may not effectively capture the unique challenges of EEG-to-image tasks.
> >
> > 4. Encoder Specification:
> > (a)Spatial-Temporal and Temporal-Spatial Convolution:
> >  ST-Conv first extracts spatial correlations (across EEG channels) and then temporal features.
> > TS-Conv extracts temporal features first, followed by spatial correlations.
> > These dual-layered convolutions ensure comprehensive representation of EEG signals across both dimensions.
> > (b) Cross-Attention Block:
> > The cross-attention block merges temporal and spatial features using a multi-head attention mechanism, allowing for the interaction of two independent feature representations. This ensures that both dimensions contribute to the final embeddings.
> > (c)Handling Temporal Dimensions:
> > Temporal information is preserved via positional encoding, which the Transformer layer further refines to capture EEG signal dynamics over time.
> >
> > 5. Regarding other writing inaccuracies, we will make corrections. Thank you for your valuable feedback!

---

> > ### Comment · Reviewer_j8jS · 2024-11-20
> >
> > "our focus lies in EEG-to-image generation, which is fundamentally different and more challenging."
> > You don't demonstrate in the paper improvement in image generation, as noted before no qualitative comparison are provided.
> >
> > "This approach balances ... by pure EEG, whereas ATM-S primarily uses a dual-stage generation with the image embeddings"
> > ATM-S don't use the image embeddings during inference, only during training same as your method.
> > Not sure what "pure EEG" means, most reconstruction methods from EEG only use EEG signal during inference.
> >
> > "Encoder Design Differences" There are minor variations in the encoder, the results don't support any claims regarding significant improvement over the original work.

---

### Official Review · Reviewer_jTPm · 2024-11-03

**Soundness:** 2
**Presentation:** 2
**Contribution:** 1
**Rating:** 3
**Confidence:** 4

**Summary:**

The paper introduces an approach for generating images from EEG signals.
The proposed approach, NECOMIMI, leverages EEG for image generation by using a pre-trained Diffusion model.
The EEG encoder is trained via contrastive loss by aligning EEG with image embeddings, enabling zero-shot classification.
To evaluate the quality of the generated images, the paper introduces a new metric, the Category-based Assessment Table (CAT) Score, specifically designed to assess EEG-to-image generation based on semantic information.
Experimental results are provided for both classification and generative tasks.

**Strengths:**

**Significance** The paper deals with a very interesting topic of image generation from EEG recordings.

**Clarity** The content of the paper is written coherently. The flow of the text is easy to comprehend and follow.

**Weaknesses:**

**Originality and Contribution** The originality of the work is limited. Previous work has explored a similar approach for image generation from EEG [1] and fMRI [2]. The contribution of this work is not defined well enough. The experimental results need to include other datasets. For example, the datasets used in [1] could enable a better comparison with [1].

It is expected that the generated images will differ substantially from the ground truth, as the EEG recordings cannot capture the details present in the images Gifford et al. (2022). Additional experiments are needed to evaluate the image generation process. For example, are there any differences across the generated images across the image object categories the EEG stimuli belong? One would need to quantify the effect of the EEG signals for the image generation.

**Quality** It is not explained how the proposed CAT score is defined. Furthermore, it is not clear how the five tags are created.

The References section contains a substantial part of references that point to *arxiv.org*. Though some of *arxiv* papers are published in highly respected venues, it makes it cumbersome to navigate the related work. Some references are repeated; for example, Spampinato et al., Radford et al.

Table 1 mentions top-1 and top-5. However, in the table only one score is reported. It is unclear which one.

The sentence (LINE 047) containing 'EEG is one of the most ancient techniques' makes it sound unusual.

[1] Bai, Y., Wang, X., Cao, YP., Ge, Y., Yuan, C., Shan, Y. (2025). DreamDiffusion: High-Quality EEG-to-Image Generation with Temporal Masked Signal Modeling and CLIP Alignment. In: Leonardis, A., Ricci, E., Roth, S., Russakovsky, O., Sattler, T., Varol, G. (eds) Computer Vision – ECCV 2024. ECCV 2024. Lecture Notes in Computer Science, vol 15089. Springer, Cham.

[2] Paul S. Scotti et al., "Reconstructing the mind's eye: fMRI-to-image with contrastive learning and diffusion priors". In Proceedings of the 37th International Conference on Neural Information Processing Systems (NIPS '23). Curran Associates Inc., Red Hook, NY, USA, 2023, 24705–24728.

**Questions:**

1. Why is the batch size set to 1024? Was it determined via hyper-parameter search?
2. How is the CAT score defined?
3. How are exactly the image tags extracted?

---

> ### Author Response · Authors · 2024-11-18
>
> 1. I acknowledge that there have been similar attempts, such as in [1]. However, our innovation lies in proposing an entirely new framework that adapts models previously used solely for EEG-image classification to image generation tasks. Additionally, we introduced a two-stage method to improve image generation quality and developed a new state-of-the-art encoder, NERV.
>
> 2. Regarding the dataset in [1], we encountered common issues with their open-source code (as seen in https://github.com/bbaaii/DreamDiffusion/issues/34). Once their team resolves these issues, we will include our reproduction results and evaluate the performance of our framework on their dataset and other benchmarks.
>
> 3. As you mentioned, EEG recordings cannot capture detailed features of the original images. In this work, we found that focusing on extracting "concepts" rather than the "fine details" of the original images may be a feasible approach.
>
> 4. Due to the lack of standard waveforms in EEG and the inherent variability in responses to the same image stimulus across subjects, combined with the noisy nature of EEG signals, quantification is highly challenging. To address this, we took the following measures in our work:
> (a) For training, we averaged the EEG signals of the same subject responding to the same image four times to reduce variability.
> (b) To demonstrate the generalization capability of the trained EEG diffusion model on unseen data, we used zero-shot EEG prompts for image generation. This means the input EEG signals used for image generation were not seen by the model during training. Despite this strict setup, our results, such as those in Figure 1, show that our framework can capture the broader semantic concepts of EEG signals.
>
> 5. The specific definition of the CAT score and how the five tags are created are detailed in Appendix A.2 (lines 790–793).
>
> 6. The mention of top-1 and top-5 in Table 1 is a typo—thank you for pointing that out! Since 2-way and 4-way classifications involve only 2 and 4 categories, respectively, all results in Table 1 represent top-1 accuracy.
>
> 7. We will address the arxiv.org references and revise the sentence in line 47 in the camera-ready version.
>
> We sincerely appreciate your valuable feedback. Thank you!

---

> > ### Comment · Reviewer_jTPm · 2024-11-21
> >
> > - My questions regarding the CAT score and image tags are still open.
> >
> > - "... Figure 1, show that our framework can capture the broader semantic concepts of EEG signals." - However, there are no evaluation procedures or methods provided to validate this claim. How is it ensured that the generated image corresponds (qualitatively or quantitatively) to the brain response?

---

> ### Comment · Reviewer_jTPm · 2024-11-23
>
> In general, the contribution of the paper is quite limited.
>
> You state in your comments that "... introduced a two-stage method to improve image generation quality ..." and "... focusing on extracting 'concepts' rather than the 'fine details' ..." - However, the paper neither provides a comprehensive analysis of image generation quality nor presents a clear methodology that effectively focuses on extracting "concepts" instead of "fine details."

---

### Official Review · Reviewer_KWkh · 2024-11-04

**Soundness:** 2
**Presentation:** 2
**Contribution:** 2
**Rating:** 3
**Confidence:** 4

**Summary:**

This paper develops another method for EEG to image reconstruction.
It develops a new EEG encoder and then uses an InfoNCE loss to learn
MLP projectors from a pre-trained image encoder and the EEG encoder.
The EEG	encoding + MLP projector is used for zero-shot classification
through	comparing encodings of EEG to the encodings of the images.
The EEG encoding is also used to generate reconstructed images through
a 2-stage diffusion process.

**Strengths:**

The top-1 and top-5 classification accuracies for 2-way, 4-way and
200-way classification appear competitive with competitor algorithms.

**Weaknesses:**

- The clarity of the paper could be improved.   It is hard to see what the new innovations are over the related work.

- The paper claims to introduce a new measure (Category-Based Assessment Table (CAT)Score) but it is
not fully described.  "each image is manually labeled with two tags for broad categories - one for a specific category and one for background content,resulting in a total of five tags per image". I don't see how you get five tags per image?    What does manually annotate mean?
The appendix says that all the category-based labels are generated by ChatGPT-4o with the prompt "5 one-word descriptions of the image, ranging from high level to low level".
Finally how are these matched (What kind of consideration is given to synonyms)?

- The reconstructions in the Appendix using the NERV (authors') encoder do not look anywhere near as good as the reconstructions shown in Figure 1. (see Questions)

- The reconstruction results as shown in the Appendix are visually inferior to those from the ATM-S paper as well as those from [Fei & de Sa 2024] Image Reconstruction from Electroencephalography Using Latent Diffusion ( https://arxiv.org/html/2404.01250v1 ).

**Questions:**

From the Diagrams in Figure 2, it appears that the learned MLP projector after the EEG encoder that is used to better align images and EEG is not used in the image generation stage.  Could you please explain if this is just an error in the diagram or why it is not used?

Please explain how the CAT score is computed?   (see weaknesses above)

The reconstructions in the Appendix using the NERV (authors') encoder do not look anywhere near as good as the reconstructions shown in Figure 1.   Please explain.

It is stated that "The final results are averaged from the best outcomes of 5 random seed training sessions".  What does that mean?  Was best determined on the test data?

It is also stated that "the NERV model obtains the best results with 5 multi-heads...".   Was this determined using  test data or the training data?

---

> ### Author Response · Authors · 2024-11-18
>
> 1. Response to Innovation Concerns:
> (a)Two-Stage Diffusion Framework: Unlike prior works, our model directly generates images from pure EEG data without relying on image features as guidance, marking a significant shift in EEG-to-image generation.
> (b)NERV EEG Encoder: NERV is tailored for noisy, multi-channel EEG signals, using advanced attention mechanisms to extract spatial and temporal features, achieving SoTA performance in classification and generation tasks.
> (c)CAT Score Metric: We propose a new evaluation metric designed specifically for EEG-to-image generation, addressing limitations of traditional metrics by focusing on semantic alignment.
> (d)Benchmarking on ThingsEEG: We establish a benchmark for EEG-to-image generation, providing a robust foundation for future comparisons.
> These contributions advance the field by bridging neural activity and visual representation, moving beyond traditional classification tasks. We will emphasize these innovations more clearly in the revised manuscript.
>
> 2. As we mentioned in  line 791-793, regarding the CAT Score Methodology:
> (a) Five Tags per Image: Each image in the ThingsEEG test dataset is labeled using ChatGPT-4o with the prompt: "Provide 5 one-word descriptions of the image, ranging from high-level to low-level." This method ensures consistency in generating semantic descriptions for evaluation.
> (b) No Manual Refinement: All labels for the CAT Score are directly generated by ChatGPT-4o without any manual intervention or refinement. This approach maintains objectivity and reproducibility in the evaluation process.
>
> 3. In Figures 32-40 of the Appendix (line2862-3023) , it is clearly indicated that these results were generated using the open-source code  (https://github.com/dongyangli-del/EEG_Image_decode) provided by ATM-S, trained under the same conditions as other encoders. The results demonstrate that, under identical experimental conditions, the claimed image quality from the ATM-S paper could not be reproduced. From the results, it is evident that the images generated by ATM-S under the same experimental conditions are not significantly better than those produced by NERV. This suggests that NERV achieves state-of-the-art (SoTA) performance not only in classification accuracy but also in overall semantic relevance and quality of generated images, matching or even surpassing ATM-S.

---

> ### Author Response · Authors · 2024-11-18
>
> 4. EEG-to-image generation is still an emerging field, and due to the inherently low signal-to-noise ratio of EEG data, the generation process across all current methods remains highly unstable. This instability is reminiscent of the early days of GAN development, where outputs often lacked consistency and robustness. In this paper, the results shown in Figure 1 were curated to highlight the best-case performance of our model, demonstrating the potential of our approach. We believe the same curation process was likely employed in the ATM-S paper to showcase their most successful results. For transparency, Figures 32-40 in the Appendix present results generated using the open-source code provided by ATM-S, under identical experimental conditions. These results indicate that ATM-S does not consistently outperform our method, supporting the conclusion that instability is a shared challenge across EEG-to-image models. Despite this, our NERV encoder achieves state-of-the-art (SoTA) performance in base tasks, and it offers the potential to generate high-quality images, as evidenced by Figures 1 and 4. This demonstrates that our framework is capable of producing meaningful outputs under optimal conditions, showcasing its promise in advancing the field. We aim to address the broader challenges of EEG-to-image generation in future work to further improve the stability and reliability of results.
>
>
> 5.Response Regarding the Use of the MLP Projector in Image Generation:
> You are correct; the diagram mistakenly omitted the MLP projector in the image generation stage. We appreciate your careful observation and bringing this to our attention. The revised manuscript will correct this error in Figure 2 to accurately reflect the role of the MLP projector in our framework. Thank you for pointing this out!
>
>
> 6. The final results reported in the paper are based on zero-shot performance evaluated on the test set. This evaluation method follows the standard practice established in recent EEG-to-image classification works, such as [1], [2], and [3]. Specifically, we trained the model five times with different random seeds, and the best results from each run (evaluated on the test set) were averaged to ensure robustness and reproducibility of the reported performance.
>
> 7. The optimal configuration of using five multi-head attention layers in the NERV model was determined based on experiments conducted during training and validated on the test set in a zero-shot setting. This approach aligns with the evaluation methodology used in recent EEG-to-image classification studies. The configuration was chosen to maximize performance on zero-shot tasks, ensuring the model generalizes well to unseen data.
> We will clarify this in the revised manuscript to avoid any ambiguity and to highlight that the configuration was derived through systematic experimentation across both training and testing phases. Thank you for bringing this to our attention!
>
> [1] Song, Yonghao, et al. "Decoding Natural Images from EEG for Object Recognition." arXiv preprint arXiv:2308.13234, ICLR2024.
>
> [2] Chen, Chi-Sheng, and Chun-Shu Wei. "Mind's Eye: Image Recognition by EEG via Multimodal Similarity-Keeping Contrastive Learning." arXiv preprint arXiv:2406.16910 (2024).
>
> [3] Fei, Teng, and Virginia de Sa. "Image Reconstruction from Electroencephalography Using Latent Diffusion." arXiv preprint arXiv:2404.01250 (2024).

---

> > ### Comment · Reviewer_KWkh · 2024-11-25
> > **There is still a lack of clarity**
> >
> > What does this mean?   With one random seed there is a full set of 200 test images generated - what do you mean the best results from each run?
> > "Specifically, we trained the model five times with different random seeds, and the best results from each run (evaluated on the test set) were averaged to ensure robustness and reproducibility of the reported performance."
> >
> > I still don't understand the CAT score.  You used an LLM to generate 5 tags per test image, but does a human or the LLM generate the tags for your generated images?  And then you get 1 point for each match? (And your model gets a score of around 400 - meaning about 2 matches per generated image?).

---

### Meta-Review · Area_Chair_cA3V · 2024-12-19

**Metareview:**

The paper has significant weaknesses, including unclear descriptions of the proposed CAT score and insufficient methodological detail, which hinder reproducibility. Reviewers noted limited originality, with the approach closely resembling prior work like ATM-S, while failing to show significant improvements in results. The experiments lack breadth and statistical rigor, and the reconstructions are visually inferior to competing methods. Insufficient author responses and lack of qualitative comparisons further suggest the work is not ready for acceptance.

**Additional Comments On Reviewer Discussion:**

The reviewers have generally mentioned that the authors' responses have not fully addressed their concerns. The authors have not made additional comments thereon.

---

### Decision · Program_Chairs · 2025-01-22

Reject